# Kernel Learning with Adversarial Features: Numerical Efficiency and Adaptive Regularization

**Antônio H. Ribeiro**
Uppsala University
antonio.horta.ribeiro@it.uu.se

**David Vävinggren**
Uppsala University
david.vavinggren@it.uu.se

**Dave Zachariah**
Uppsala University
dave.zachariah@it.uu.se

**Thomas B. Schön**
Uppsala University
thomas.schon@uu.se

**Francis Bach**
PSL Research University / INRIA
francis.bach@inria.fr

## Abstract

Adversarial training has emerged as a key technique to enhance model robustness against adversarial input perturbations. Many of the existing methods rely on computationally expensive min-max problems that limit their application in practice. We propose a novel formulation of adversarial training in reproducing kernel Hilbert spaces, shifting from input to feature-space perturbations. This reformulation enables the exact solution of inner maximization and efficient optimization. It also provides a regularized estimator that naturally adapts to the noise level and the smoothness of the underlying function. We establish conditions under which the feature-perturbed formulation is a relaxation of the original problem and propose an efficient optimization algorithm based on iterative kernel ridge regression. We provide generalization bounds that help to understand the properties of the method. We also extend the formulation to multiple kernel learning. Empirical evaluation shows good performance in both clean and adversarial settings.

## 1 Introduction

Adversarial training can be used to learn models that are robust against input perturbations (Madry et al., 2018). It considers training samples that have been modified by an adversary, with the goal of obtaining a model that will be more robust when faced with newly perturbed samples. Consider a training dataset $\{(\boldsymbol{x}_i, y_i)\}_{i=1}^n$ consisting of $n$ samples of dimension $\mathcal{X} \times \mathbb{R}$. The training procedure is formulated as the following min-max optimization problem:

$$\min_{\boldsymbol{f} \in \mathcal{H}} \frac{1}{n} \sum_{i=1}^{n} \max_{\Delta \boldsymbol{x}_i \in \Omega_{\mathcal{X}}} \ell(y_i, \boldsymbol{f}(\boldsymbol{x}_i + \Delta \boldsymbol{x}_i)), \tag{1}$$

finding the best function $\boldsymbol{f} \in \mathcal{H}$ subject to the worst disturbances in a predefined set of admissible input perturbations $\Omega_{\mathcal{X}}$, for instance, $\Omega_{\mathcal{X}} = \{\Delta \boldsymbol{x} : \|\Delta \boldsymbol{x}\| \leq \delta\}$. For nonlinear functions, solving this problem directly is often challenging; simple optimization typically requires two stages. First we differentiate through the inner maximization, which is often approximated via projected gradient descent (Madry et al., 2018), and then we differentiate again through the solver steps in the computation of gradients through the outer minimization.

We consider $\mathcal{H}$ to be a reproducing kernel Hilbert space (RKHS) and represent the function $\boldsymbol{f}$ as a linear model applied to a (potentially infinite-dimensional) feature transformation of the input. Let $\phi(\boldsymbol{x})$ denote the corresponding feature map, so that any $\boldsymbol{f} \in \mathcal{H}$ evaluated at $\boldsymbol{x}$ can be expressed as $\boldsymbol{f}(\boldsymbol{x}) = \langle \boldsymbol{f}, \phi(\boldsymbol{x}) \rangle$, where $\langle \cdot, \cdot \rangle$ denotes the inner product in $\mathcal{H}$, see Schölkopf and Smola (2002). In this paper, we introduce perturbations directly in the feature space rather than in the input space.

39th Conference on Neural Information Processing Systems (NeurIPS 2025).

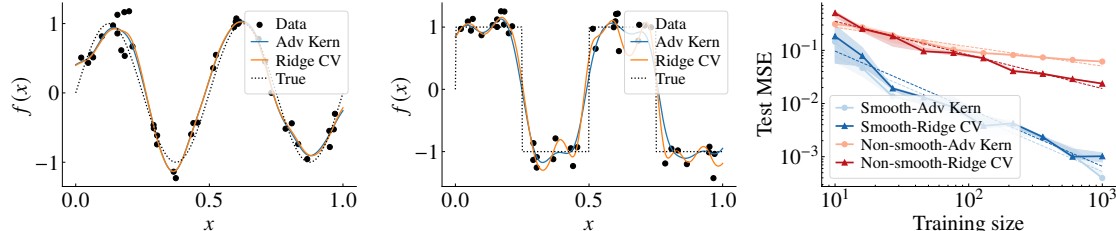

Figure 1: We compare adversarial kernel training with cross-validated kernel ridge regression (Ridge CV). All experiments make use of the Matern-$5/2$ kernel. *Left:* we show how it fits a smooth target function. *Middle:* a non-smooth target function. *Right:* we show the test mean square error (MSE) vs the size of the training dataset $n$, and illustrate how adversarial kernel training adapts to the target function similarly to kernel ridge regression with cross validation, without the need of tuning on hyperparameters.

This leads to the following formulation:

$$\min_{\boldsymbol{f} \in \mathcal{H}} \frac{1}{n} \sum_{i=1}^{n} \max_{\boldsymbol{d} \in \Omega_{\mathcal{H}}} \ell(y_i, \langle \boldsymbol{f}, \boldsymbol{\phi}(\boldsymbol{x}_i) + \boldsymbol{d} \rangle). \tag{2}$$

Note that the disturbance is applied in a (potentially infinite-dimensional) feature space $\mathcal{H}$, rather than the input space $\mathcal{X}$. As we will show, this formulation of adversarial training is particularly amenable to both analysis and efficient optimization. Working in the feature space allows for the exact solution of inner maximization problem, allowing for more efficient optimization. We further develop a dedicated solver tailored to this formulation. Beyond its computational advantages, we establish theoretical guarantees showing that the feature-perturbed objective in Eq. (2) upper bounds its input-perturbed counterpart in Eq. (1), ensuring that optimizing the former provides robustness guarantees for the latter. Empirically, we also demonstrate that this formulation achieves strong robustness to adversarial input perturbations.

We study the properties of our method not only in **adversarial**, but also in **clean data**, analysing the **regularization properties** of our estimator. We are interested in this estimator promoting model simplicity and enhancing generalization to unseen, non-adversarial data. In the linear (finite-dimensional) case, adversarial training has been shown to behave similarly to parameter-shrinking approaches such as lasso and ridge regression (Ribeiro et al., 2023). In this setting, unlike when we are trying to defend against a threat model with fixed adversarial radius $\delta$, we can allow the training adversarial radius to decay as we get more data. Particularly, if we allow $\delta \propto 1/\sqrt{n}$, our estimator is consistent and also have near-oracle performance (Ribeiro et al., 2023; Xie & Huo, 2024). Previous work argue that a key advantage of adversarial training as an alternative to parameter shrinking methods is that it achieves the same sample-efficiency rates as the latter without requiring $\delta$ to depend on the noise level (Ribeiro et al., 2023; Xie & Huo, 2024), paralleling the square-root Lasso (Belloni et al., 2011). In this paper, we prove that similar properties hold for our infinite-dimensional formulation in (2), with default $\delta$ set without requiring knowledge of the signal-to-noise ratio yielding similar rates to kernel regression when this quantity is known. In practice, as shown in Figure 1, the method has an adaptive regularization property with the default choice often performing comparably with kernel ridge regression with a parameter chosen using cross-validation. The rightmost plot further highlights the method's adaptivity to the underlying function class.

**Our contributions** are to propose and analyze feature-perturbed adversarial kernel training. We:

- **Establish equivalences between input and feature-perturbed formulation:** We identify conditions under which the feature-perturbed formulation (2) upper-bounds the input-perturbed objective (1), ensuring that solving the former yields guarantees for the latter.
- **Propose an efficient solver:** We show that the feature-perturbed problem (2) allows for the inner maximization to be solved exactly and derive a tailored solver for its optimization.
- **Provide generalization guarantees:** We derive upper bounds for the generalization error that highlight how the hyperparameter can be set without access to noise levels, leading to practical performance without hyperparameter tuning, with natural adaptivity to noise levels.
- **Extend the framework to multiple kernel learning:** We generalize our formulation to support combinations of kernels, enhancing expressivity.

These results demonstrate the practical and theoretical appeal of feature-perturbed adversarial learning. We validate the method on real and simulated data. We make our implementation available at `github.com/antonior92/adversarial_training_kernel`.

Table 1: Kernel functions and choices of $\Omega_{\mathcal{X}}$ for which $\Omega_{\mathcal{X}} \subseteq \{\Delta x : D_{\mathcal{H}}(x, x + \Delta x) \leq \delta\}$. Equivalences proved in Appendix A. For the polynomial kernel, the constant $C$ depends on $\|x\|$ and $\|x + \Delta x\|$ and is specified in the appendix.

| Kernel | $k(x, x')$ | $\Omega_{\mathcal{X}}$ |
|--------|-----------|------------------------|
| **Linear** | $x^\top x'$ | $\|\Delta x\|_2 \leq \delta$ |
| **Polynomial** | $(1 + x^\top x')^d$ | $\|\Delta x\|_2 \leq C\delta^{1/d}$ |
| **Exponential** | $\exp(-\gamma\|x - x'\|_2)$ | $\|\Delta x\|_2 \leq \frac{1}{\gamma}\log\frac{2}{2-\delta}$ |
| **Gaussian** | $\exp(-\gamma\|x - x'\|_2^2)$ | $\|\Delta x\|_2 \leq \sqrt{\frac{1}{\gamma}\log\frac{2}{2-\delta}}$ |
| **Laplacian** | $\exp(-\gamma\|x - x'\|_1)$ | $\|\Delta x\|_1 \leq \frac{1}{\gamma}\log\frac{2}{2-\delta}$ |

## 2  Background

Different estimators will be relevant to our developments and will be compared in this paper. Let $\mathcal{H}$ be a Reproducing Kernel Hilbert Space (RKHS) associated with the kernel $k(x, x') = \langle \phi(x), \phi(x') \rangle$ and the induced norm $\|\cdot\|_{\mathcal{H}}$. See Bach (2024) and Schölkopf and Smola (2002). Here, we will **focus on the squared loss** for simplicity, but some of the developments can apply more generally.

**Adversarial kernel training** is our main object of study. We will call the following estimator *feature-perturbed adversarial kernel training*:

$$\min_{f \in \mathcal{H}} \frac{1}{n} \sum_{i=1}^{n} \max_{d \in \Omega_{\mathcal{H}}} (y_i - \langle f, \phi(x_i) + d \rangle)^2,$$

where $\Omega_{\mathcal{H}} = \{d : \|d\|_{\mathcal{H}} \leq \delta\}$. This estimator stands in contrast to the *input-perturbed adversarial kernel training* defined in Equation (1), where the perturbation is applied to the input rather than the features. For simplicity, and unless otherwise specified, we use the term *adversarial kernel training* to refer to the feature-perturbed variant, which is the main focus of this work. The feature-perturbed formulation is a relaxation of the input-perturbed counterpart, as it will be explained in Section 3. We will also show that it is closely related to kernel ridge regression.

**Kernel ridge regression** (Bach, 2024; Schölkopf & Smola, 2002) estimates $f \in \mathcal{H}$ by solving,

$$\min_{f \in \mathcal{H}} \frac{1}{n} \sum_{i=1}^{n} (y_i - f(x_i))^2 + \lambda\|f\|_{\mathcal{H}}^2, \tag{3}$$

where $\mathcal{H}$ is a reproducing kernel Hilbert space. Here, even though we might be optimizing over an infinite-dimensional space, the kernel trick can be used to transform the problem into a regularized least squares problem using a kernel matrix.

**Multiple kernel learning** (Lanckriet et al., 2004; Rakotomamonjy et al., 2008) consists of finding a set of functions $\{f_j \in \mathcal{H}_j, j = 1, \cdots, p\}$ by solving

$$\min_{\substack{f_j \in \mathcal{H}_j \\ j=1,\dots,p}} \frac{1}{n} \sum_{i=1}^{n} (y_i - \sum_{j=1}^{p} f_j(x_i))^2 + \lambda\Big(\sum_{j=1}^{p} \|f_j\|_{\mathcal{H}_j}\Big)^2. \tag{4}$$

The framework combines multiple kernel functions—each representing a different notion of similarity between data points—into a single model, allowing the algorithm to simultaneously learn the model parameters and the best combination of kernels. This is particularly useful when data may have multiple modalities, or when it is not possible for a single kernel to capture all relevant patterns in the data. In Section 6 we propose an *adversarial multiple kernel learning* method.

## 3  Problem formulation

Here, we will establish properties of adversarial kernel training. Particularly, we analyse the feature-perturbed adversarial error. We first prove that it upper bounds the input-perturbed adversarial error and later show that it has an easy-to-analyze closed-form solution.

**Relation between feature and input perturbations:** In the case of RKHS, we can always find $\Omega_{\mathcal{X}}$ for which feature-perturbed adversarial kernel training is a relaxation of the input-perturbed adversarial kernel training, with Eq. (2) being an upper bound to Eq. (1). Particularly, the result holds for inputs within a ball in the kernel distance $D_{\mathcal{H}}(\boldsymbol{x}, \boldsymbol{x}') = \|\phi(\boldsymbol{x}) - \phi(\boldsymbol{x}')\|_{\mathcal{H}}$, as stated in the next proposition (proved in the appendix). Table 1 also provides concrete examples of a few kernels and the sets $\Omega_{\mathcal{X}}$ for which the above proposition holds. The proof is provided in Appendix A.1.

---

**Proposition 1.** Let $\Omega_{\mathcal{H}} = \{\boldsymbol{d} : \|\boldsymbol{d}\|_{\mathcal{H}} \leq \delta\}$ and $\Omega_{\mathcal{X}} = \{\Delta\boldsymbol{x} : D_{\mathcal{H}}(\boldsymbol{x}, \boldsymbol{x} + \Delta\boldsymbol{x}) \leq \delta\}$ then:

$$\max_{\boldsymbol{d}\in\Omega_{\mathcal{H}}} (y - \langle \boldsymbol{f}, \phi(\boldsymbol{x}) + \boldsymbol{d}\rangle)^2 \geq \max_{\Delta\boldsymbol{x}\in\Omega_{\mathcal{X}}} (y - \boldsymbol{f}(\boldsymbol{x} + \Delta\boldsymbol{x}))^2, \text{ for all } \boldsymbol{f} \in \mathcal{H}.$$

---

**The dual formulation of feature-perturbed adversarial error:** Feature perturbed adversarial training is closely related to kernel ridge regression. The following proposition extends earlier work (Ribeiro et al., 2023) and provides a characterization that makes this connection explicit.

---

**Proposition 2** (closed formula for inner-maximization)**.** If $\Omega_{\mathcal{H}} = \{\boldsymbol{d} : \|\boldsymbol{d}\|_{\mathcal{H}} \leq \delta\}$ for every $\boldsymbol{x}, y, \boldsymbol{f}$, then we have:

$$\max_{\boldsymbol{d}\in\Omega_{\mathcal{H}}} (y - \langle \boldsymbol{f}, \phi(\boldsymbol{x}) + \boldsymbol{d}\rangle)^2 = (|y - \boldsymbol{f}(\boldsymbol{x})| + \delta\|\boldsymbol{f}\|_{\mathcal{H}})^2.$$

---

This result shows that adversarial training loss under RKHS-norm bounded perturbations admits a closed-form expression. As a consequence, our original optimization problem in Eq. (2) can be equivalently rewritten as:

$$\min_{\boldsymbol{f}\in\mathcal{H}} \frac{1}{n} \sum_{i=1}^{n} (|y_i - \boldsymbol{f}(\boldsymbol{x}_i)| + \delta\|\boldsymbol{f}\|_{\mathcal{H}})^2, \tag{5}$$

which by inspection is very similar to ridge regression (for which the norm is outside the parentheses), and suggests similar properties. Still, as we will see in Section 5, the two problems have important differences that might make it advantageous to use the latter formulation rather than kernel ridge regression.

This reformulation highlights an appealing property of feature-perturbed adversarial training. While both the input-perturbed and feature-perturbed formulations can be expressed as pointwise maxima of convex functions (and are therefore convex), the inner maximization in the input-perturbed case is non-concave and generally intractable. In contrast, the feature-perturbed formulation admits the closed-form solution derived above, yielding a convex and computationally favorable objective that preserves a clear interpretation in terms of robustness to RKHS-bounded perturbations. In the following, we propose an iterative algorithm to efficiently solve this problem.

## 4 Optimization

In this section, we propose an iterative kernel ridge regression algorithm (Algorithm 1) for solving (2) efficiently. The algorithm proposed here is influenced by Ribeiro et al. (2025), but adapted to non-parametric regression from the linear regression setting. Below, we describe the key parts of the algorithm, while additional details are available in the appendix.

### 4.1 Proposed algorithm

Algorithm 1 follows from the idea of using the following variational reformulation of the loss:

$$(|y - \boldsymbol{f}(\boldsymbol{x})| + \delta\|\boldsymbol{f}\|_{\mathcal{H}})^2 = \inf_{\eta^0,\eta^1} \frac{|y - \boldsymbol{f}(\boldsymbol{x})|^2}{\eta^0} + \frac{\delta^2\|\boldsymbol{f}\|_{\mathcal{H}}^2}{\eta^1} \tag{6}$$

$$\text{s.t. } \eta^0 + \eta^1 = 1, \eta^0 > 0, \eta^1 > 0.$$

The above reformulation is often referred to as $\eta$-trick (Bach, 2011, 2019). And, for $|y - \boldsymbol{f}(\boldsymbol{x})| > 0$ and $\|\boldsymbol{f}\|_{\mathcal{H}} > 0$, allows the closed formula solution:

$$\eta^0 = \frac{|y - \boldsymbol{f}(\boldsymbol{x})| + \delta\|\boldsymbol{f}\|_{\mathcal{H}}}{|y - \boldsymbol{f}(\boldsymbol{x})|}, \quad \eta^1 = \frac{|y - \boldsymbol{f}(\boldsymbol{x})| + \delta\|\boldsymbol{f}\|_{\mathcal{H}}}{\delta\|\boldsymbol{f}\|_{\mathcal{H}}}. \tag{7}$$

We can use this trick to rewrite the original loss as the squared sum as a sum of squares. We apply the equivalence in (6) individually to each sample, yielding:

---

**Algorithm 1** Iterative Kernel Ridge Regression

---

**Initialize:** weights $w_i \leftarrow 1, i = 1, \ldots, n$; and, $\lambda \leftarrow \delta$
**Repeat:**
1. *Solve* reweighted kernel ridge regression:

$$\widehat{\boldsymbol{f}} \leftarrow \arg\min_{\boldsymbol{f}} \frac{1}{n} \sum_{i=1}^{n} w_i (y_i - \boldsymbol{f}(\boldsymbol{x}_i))^2 + \lambda \|\boldsymbol{f}\|_{\mathcal{H}}^2$$

2. *Update* weights (using Eq. (9)): $\boldsymbol{w}, \boldsymbol{\lambda} \leftarrow \texttt{UpdateWeights}(\widehat{\boldsymbol{f}})$
3. *Quit* if $\texttt{StopCriteria}$.

---

$$\min_{\boldsymbol{f}} \inf_{\eta_i^0, \eta_i^1} \frac{1}{n} \sum_{i=1}^{n} \frac{|y_i - \boldsymbol{f}(\boldsymbol{x}_i)|^2}{\eta_i^0} + \frac{1}{n} \sum_{i=1}^{n} \frac{\delta^2}{\eta_i^1} \|\boldsymbol{f}\|_{\mathcal{H}}^2 \tag{8}$$
$$\text{s.t. } \eta_i^0 + \eta_i^1 = 1, \eta_i^0 > 0, \eta_i^1 > 0, \forall i.$$

We solve it as a block-wise coordinate problem, alternating between 1) solving for $\eta^0, \eta^1$ and computing the weights

$$w_i = \frac{1}{\eta_i^0}, \quad \lambda = \frac{1}{n} \sum_i \frac{\delta^2}{\eta_i^1} \text{for all } i \tag{9}$$

and 2) solving the kernel ridge regression problem that arises to find an approximated $\boldsymbol{f}$.

### 4.2 Additional analysis and comments

**Practical implementation.** To make the algorithm numerically stable, we use a variation of the $\eta$-trick with an added $\epsilon$ that guarantees convergence to the solution of (8) as $\epsilon \to 0$. See Bach (2019) and Ribeiro et al. (2025). Using $\epsilon > 0$ is important to avoid numerical problems and to guarantee convergence. Otherwise, when either $|y - \boldsymbol{f}(\boldsymbol{x})|$ or $\|\boldsymbol{f}\|_{\mathcal{H}}$ are too small, we could obtain extremely large values for $1/\eta$ and an ill-conditioned problem.

**Convergence.** The convergence of the above algorithm (for $\epsilon > 0$) follows from results about the convergence of block coordinate descent for problems that are jointly convex and differentiable with unique solutions along each direction (Luenberger & Ye, 2008, Section 8.6).

**Computational complexity.** Solving kernel ridge regression typically incurs a computational cost of $O(n^3)$ due to the need to factor the kernel matrix. This term usually dominates the cost of Algorithm 1, that usually converges within a few iterations (Bach, 2019; Ribeiro et al., 2025). To further reduce complexity, Ribeiro et al. (2025) proposes using the conjugate gradient (CG) method to approximately solve the reweighted ridge regression problem with only quadratic complexity. Similar considerations apply to our setting. Moreover, since our formulation relies on kernel evaluations, it is naturally compatible with Nyström approximations (Williams & Seeger, 2001), offering additional scalability for large datasets. While these extensions are promising, they fall outside our scope and are not explored in the current study.

## 5 Generalization bounds

We establish worst-case error bounds for adversarial kernel training. Particularly, we assume that the data is generated as

$$y = \boldsymbol{f}^*(\boldsymbol{x}) + \sigma \boldsymbol{w},$$

where $\boldsymbol{f}^*$ is the function that we want to recover and $\boldsymbol{w}$ is a unitary vector representing the noise. We use $\sigma$ to quantify the magnitude of the noise, and $R$ to denote the function magnitude $R = \|\boldsymbol{f}^*\|_{\mathcal{H}}$. We provide upper bounds on the in-sample excess risk:[1]

$$\mathcal{E}(\boldsymbol{f} - \boldsymbol{f}^*) = \frac{1}{n} \sum_{i=1}^{n} (\boldsymbol{f}(\boldsymbol{x}_i) - \boldsymbol{f}^*(\boldsymbol{x}_i))^2.$$

This setting is also called fixed-design setting (Bach, 2024). We choose to study it because it provides maximum intuition without some of the additional technical challenges of the random-design

---

[1]This quantity is called excess risk because it is equal to the difference between the expected risk $\mathbb{E}_{\boldsymbol{w}}[\frac{1}{n} \sum_{i=1}^{n} (\boldsymbol{f}(\boldsymbol{x}_i) - y_i)^2]$ and the Bayes optimal risk $\mathbb{E}_{\boldsymbol{w}}[\frac{1}{n} \sum_{i=1}^{n} (\boldsymbol{f}^*(\boldsymbol{x}_i) - y_i)^2]$ (Bach, 2024, Prop. 3.3).

case, where we would bound $\mathbb{E}_{\boldsymbol{x}}[\boldsymbol{f}(\boldsymbol{x}) - \boldsymbol{f}^*(\boldsymbol{x})]$. As discussed Section 5.3, bounds in the fixed-design setting can also inform results in the random-design setting (Wainwright, 2019, Chap. 14). As we will see, adversarial training has the desirable property that the parameter $\delta$ can be set without knowledge of the noise magnitude $\sigma$. The same is not true for kernel ridge regression.

Our analysis is as follows: we first consider deterministic values of $\boldsymbol{w}$ and the function $\boldsymbol{f}^*$ within the model class, that is, $\boldsymbol{f}^* \in \mathcal{H}$. We define $\gamma$ and $\beta \in \mathbb{R}$ as follows:

$$\gamma = \sup_{\|\boldsymbol{g}\|_{\mathcal{H}} \leq 1} \frac{1}{n} \sum_{i=1}^{n} w_i \boldsymbol{g}(\boldsymbol{x}_i), \quad \beta = \sup_{\mathcal{E}(\boldsymbol{g}) \leq 1} \frac{1}{n} \sum_{i=1}^{n} w_i \boldsymbol{g}(\boldsymbol{x}_i), \tag{10}$$

where we take supremum over all $\boldsymbol{g} \in \mathcal{H}$, and we obtain bounds in terms of this quantities. This analysis is described in Section 5.1.

Next, we obtain high probability bounds for the case where the noise is Gaussian. Particularly under Gaussian noise we have, with high probability, that $\gamma = O(n^{-1/2})$ **for linear or any translational invariant kernel**. On the other hand, with high probability, $\beta^2 = O(p/n)$ **for linear kernel** or **for the matérn kernel** $\beta^2 = O(n^{-\frac{2}{2+p/\nu}})$. We will discuss these results in detail in Section 5.2. Finally, Section 5.3 mentions generalization to the case where the model is mispecified or to the random design setting. This progression allows us to progressively analyse more advanced cases, with each section building on the previous one.

## 5.1 Deterministic bounds and well-specified models

Let consider deterministic values of $\boldsymbol{w}$, chosen such that $\frac{1}{n} \sum_{i=1}^{n} w_i^2 = 1$. The next theorem bounds the excess risk in terms of $\gamma$ and $\beta$. We are using proof techniques similar to those in Wainwright (2019, Chapters 7 and 13), and also related to those developed in the context of adversarial training (Ribeiro et al., 2023; Xie & Huo, 2024).

---

**Theorem 1** (Excess risk for adversarial kernel training). Let $\sigma$ quantify the magnitude of the noise, and $R = \|\boldsymbol{f}^*\|_{\mathcal{H}}$ denote the function magnitude, and $\delta$ be the adversarial training radius. We have the following upper bound on the excess risk:

$$\mathcal{E}(\widehat{\boldsymbol{f}} - \boldsymbol{f}^*) \leq \min(\mathcal{B}_{\gamma}^{\mathrm{adv}}, \mathcal{B}_{\beta}^{\mathrm{adv}}).$$

where $\mathcal{B}_{\gamma}^{\mathrm{adv}} = 4(\frac{2\sigma R}{\delta} + R^2)(\delta + \gamma)^2$ and $\mathcal{B}_{\beta}^{\mathrm{adv}} = 4\sigma\delta R + 10\delta^2 R^2 + 16\sigma^2\beta^2$.

---

Considering $\mathcal{B}_{\gamma}^{\mathrm{adv}}$ and $\mathcal{B}_{\beta}^{\mathrm{adv}}$ allows us to describe two different regimes. The optimal choice for $\mathcal{B}_{\gamma}^{\mathrm{adv}}$ is to set $\delta \propto \gamma$. In this case, ignoring constants and higher order terms: $\mathcal{B}_{\gamma}^{\mathrm{adv}} = O(\sigma R\gamma)$. The bound based on $\mathcal{B}_{\beta}^{\mathrm{adv}}$ is a bound that does not really improve as we increase $\delta$, and the optimal choice here would be $\delta = 0$. Overall, for any $\delta < \frac{\sigma}{R}\beta^2$, we obtain $\mathcal{B}_{\beta}^{\mathrm{adv}} = O(\sigma^2\beta^2)$. Next, we present the equivalent result for kernel ridge regression for comparison.

---

**Theorem 2** (Excess risk for kernel ridge regression). Let $\sigma$ quantify the magnitude of the noise, $R = \|\boldsymbol{f}^*\|_{\mathcal{H}}$, and let $\lambda$ be the regularization parameter in kernel ridge regression Equation (3). We have the following upper-bound on the excess risk $\mathcal{E}(\widehat{\boldsymbol{f}} - \boldsymbol{f}^*) \leq \min(\mathcal{B}_{\gamma}^{\mathrm{kr}}, \mathcal{B}_{\beta}^{\mathrm{kr}})$. Where $\mathcal{B}_{\gamma}^{\mathrm{kr}} = 2\frac{(R\lambda + \sigma\gamma)^2}{\lambda}$ and $\mathcal{B}_{\beta}^{\mathrm{kr}} = 2(R^2\lambda + \sigma^2\beta^2)$.

---

Similar considerations apply here and each bound allows us to describe a different regime. The optimal choice for $\lambda$ optimizing $\mathcal{B}_{\gamma}^{\mathrm{kr}}$ is to set $\lambda \propto \frac{\sigma}{R}\gamma$. In this case, $\mathcal{B}_{\gamma}^{\mathrm{kr}} = O(\sigma R\gamma)$. The bound based on $\mathcal{B}_{\beta}^{\mathrm{kr}}$ is a bound that does not really improve as we increase $\lambda$, and the optimal choice here would be $\lambda = 0$. Overall, for any $\lambda < \frac{\sigma^2}{R^2}\beta^2$, we obtain $\mathcal{B}_{\beta}^{\mathrm{kr}} = O(\sigma^2\beta^2)$.

**Signal-to-noise ratio.** Notice that for kernel ridge regression if we set $\lambda$ without knowledge of the signal-to-noise ratio $\sigma/R$, for instance if $\lambda \propto \gamma$, the resulting excess risk is: $\mathcal{B}_{\gamma}^{\mathrm{kr}} = O((\sigma^2 + R^2)\gamma)$. Indeed, kernel ridge regression exhibits a quadratic dependency on the noise level unless $\lambda$ is tuned with prior knowledge of the signal-to-noise ratio. In contrast, adversarial kernel training adapts to the noise level automatically, achieving near-optimal performance without requiring knowledge of $\sigma/R$. This adaptivity is a key advantage of the latter approach.

## 5.2 High-probability bounds under Gaussian noise

In this subsection, we assume the noise variables are Gaussian and i.i.d. $\boldsymbol{w} \sim N(0, I_n)$. We define $\bar{\gamma}$ and $\bar{\beta}$ as follows:

$$\bar{\gamma} = \mathbb{E}_{\boldsymbol{w} \sim N(0,I_n)} \left[ \sup_{\|\boldsymbol{g}\|_{\mathcal{H}} \leq 1} \frac{1}{n} \sum_{i=1}^{n} w_i(\boldsymbol{g}(\boldsymbol{x}_i)) \right], \quad \bar{\beta}^2 \geq \mathbb{E}_{\boldsymbol{w} \sim N(0,I_n)} \left[ \sup_{\mathcal{E}(\boldsymbol{g}) \leq \bar{\beta}^2} \frac{1}{n} \sum_{i=1}^{n} w_i(\boldsymbol{g}(\boldsymbol{x}_i)) \right]. \quad (11)$$

These are usually called Gaussian complexity and local Gaussian complexity. Where we consider any value $\bar{\beta}$ satisfying the inequality. The following results allow us to use $\bar{\gamma}$ and $\bar{\beta}$ in our analysis, showing that they upper bound $\gamma$ and $\beta$ with high probability.

> **Proposition 3.** Let $\mu_1 > \mu_2 > \cdots > \mu_n$ be the eigenvalues of the kernel matrix $K$, i.e., the matrix with entries $K_{i,j} = k(\boldsymbol{x}_i, \boldsymbol{x}_j)$. If $\boldsymbol{w} \sim N(0, I_n)$ and $\epsilon > 0$, then:
>
> - We have $\gamma \leq \bar{\gamma} + \epsilon$ with probability higher than $1 - e^{-\frac{n^2 \epsilon}{2\lambda_1}}$;
> - And, $\beta \leq \bar{\beta} + \epsilon$ with probability $1 - \exp(-\frac{n(\bar{\beta}+\epsilon)}{2})$, as long as $\sqrt{\mathcal{E}(\widehat{\boldsymbol{f}} - \boldsymbol{f}^*)} \geq \bar{\beta} + \epsilon$.

We can obtain closed-form expressions for $\bar{\gamma}$ and $\bar{\beta}^2$, which enable a more explicit analysis of the upper bounds derived in the previous section. We do it for adversarial training, but similar argument holds for kernel ridge regression.

**On the dimension-free bounds depending on $\bar{\gamma}$.** We have $\bar{\gamma} = \frac{\sqrt{\mathrm{tr}(K)}}{n}$, where $\mathrm{tr}(\cdot)$ denotes the matrix trace. For any normalized kernel, i.e., one satisfying $k(\boldsymbol{x}, \boldsymbol{x}) = 1$, it follows that $\mathrm{tr}(K) = n$ and thus $\bar{\gamma} = 1/\sqrt{n}$. Examples include Gaussian and Matérn kernels. A similar rate holds for the linear kernel: if $M = \max_i \|\boldsymbol{x}_i\|_2$, then $\bar{\gamma} = M/\sqrt{n}$. Combining this with the results from the previous section, we obtain the following dimension-free bound for adversarial training:

$$\mathcal{B}_{\gamma}^{\mathrm{adv}} = O\left( \frac{\sigma R}{\sqrt{n}} \right) \qquad \text{for} \qquad \delta \propto \frac{1}{\sqrt{n}} \qquad \textbf{(Linear or translation-invariant kernel)}$$

We refer to this as a *dimension-free bound* since it does not depend explicitly on the input dimension $p$, although $p$ may still influence the $\sigma$, $R$, or $M$.

**On the faster (but dimension-dependent) rates in $\bar{\beta}^2$.** We can also derive closed-form expressions for $\bar{\beta}$. The derivations are very similar to that of (Wainwright, 2019, Chapter 13) and we provide aditional details in Appendix C. In particular, we have $\bar{\beta}^2 = O(p/n)$ for the linear kernel and $\bar{\beta}^2 = O(n^{-2/(2+p/\nu)})$ for the Matérn kernel. This leads to the following bounds:

$$\mathcal{B}_{\beta}^{\mathrm{adv}} = O\left( \frac{\sigma^2 p}{n} \right) \qquad \text{for} \qquad \delta < \frac{\sigma}{R} \frac{p}{n} \qquad \textbf{(Linear kernel)}$$

and

$$\mathcal{B}_{\beta}^{\mathrm{adv}} = O\left( \sigma^2 n^{-2/(2+p/\nu)} \right) \qquad \text{for} \qquad \delta < \frac{\sigma}{R} n^{-2/(2+p/\nu)} \qquad \textbf{($\nu$-Matérn kernel)}$$

Here, $\delta$ is not required to scale with $n$ in a specific way—it only needs to be sufficiently small. The linear case recovers the classical $p/n$ rate of standard linear regression, which converges faster but explicitly depends on the number of features $p$.

## 5.3 Additional analysis and comments

**Bounds for the mispecified case.** For adversarial kernel regression, i.e., when $\boldsymbol{f}^* \notin \mathcal{H}$ we can use the following equations

$$\mathcal{E}(\widehat{\boldsymbol{f}} - \boldsymbol{f}^*) \leq 2 \inf_{\|\boldsymbol{f}\|_{\mathcal{H}} \leq R} \mathcal{E}(\boldsymbol{f} - \boldsymbol{f}^*) + 2 \min(\mathcal{B}_{\gamma}^{\mathrm{adv}}, \mathcal{B}_{\beta}^{\mathrm{adv}}).$$

Note that the results involve the same terms $(\mathcal{B}_{\gamma}^{\mathrm{adv}}, \mathcal{B}_{\beta}^{\mathrm{adv}})$ that appear in the well-specified case. Similar expressions also apply to kernel ridge regression. The first term in the equation is the approximation error of trying to approximate $\|\boldsymbol{f}\|_{\mathcal{H}} \leq R$ and the second term is the estimation error. For translational invariant kernels, we can leverage results from Bach (2024, Section 7.5.2) to analyse the left-hand side above.

**Bounds for the random design case.** We are often interested in the random design case, i.e., bounding $\mathbb{E}_{\boldsymbol{x}}[\boldsymbol{f}(\boldsymbol{x}) - \boldsymbol{f}^*(\boldsymbol{x})]$. We suggest that the difference between this quantity and the excess

risk $\mathcal{E}(\hat{\boldsymbol{f}} - \boldsymbol{f}^*)$ can be bounded by the Radamacher complexity (Wainwright, 2019, Chapter 14) or localized Radamacher complexity (Wainwright, 2019, Chapter 14). For instance, see Corollary 14.15, where the authors establish such results for Kernel Ridge Regression. We believe this deserves more careful investigation, but due to the relation between Gaussian and Radamacher complexity in many cases, the rate for the random design case should be the same as the ones we observe here.

# 6 Adversarial multiple kernel learning

Let us denote the space $\bar{\mathcal{H}} = \bigoplus_{j=1}^{D} \mathcal{H}_j = \{\sum_{j=1}^{D} \boldsymbol{f}_j(\boldsymbol{x}) \text{ where } \boldsymbol{f}_j \in \mathcal{H}_j \text{ for } j = 1, \cdots, D\}$. Here, we propose an adversarial multiple-kernel learning method, where we solve:

$$\min_{\boldsymbol{f} \in \bar{\mathcal{H}}} \frac{1}{n} \sum_{i=1}^{n} \max_{d \in \Omega_{\bar{\mathcal{H}}}} (y - \langle \boldsymbol{f}, \boldsymbol{\phi}(\boldsymbol{x}) + \boldsymbol{d} \rangle)^2. \tag{12}$$

We consider $\Omega_{\bar{\mathcal{H}}}$ to be the intersection of balls in different kernel spaces as attack region:

$$\Omega_{\bar{\mathcal{H}}} = \cap_{j=1}^{D} \Omega_{\mathcal{H}_j} = \cap_{j=1}^{D} \left\{ \boldsymbol{d}_j \in \mathcal{H}_j : \|\boldsymbol{d}_j\|_{\mathcal{H}_j} \leq \delta \right\} = \left\{ \boldsymbol{d} \in \cap_{j=1}^{D} \mathcal{H}_j : \max_{j=1,\cdots,D} \|\boldsymbol{d}\|_{\mathcal{H}_j} \leq \delta \right\}.$$

The framework combines multiple kernel functions—each representing a different notion of similarity between data points—into a single model, allowing the algorithm to learn both the model parameters and the best combination of kernels simultaneously. This is particularly useful when data may have multiple views or modalities, or when no single kernel captures all relevant patterns in the data. Notably, many of the techniques observed in the context of feature-perturbed adversarial kernel training have clear parallels in Equation (12). We highlight some of these connections below.

**Relation with attacks applied to the input domain.** Our optimization problem (12) is a relaxation of

$$\min_{\boldsymbol{f} \in \bar{\mathcal{H}}} \frac{1}{n} \sum_{i=1}^{n} \max_{\Delta \boldsymbol{x}_i \in \Omega_{\mathcal{X}}} (y - \boldsymbol{f}(\boldsymbol{x} + \Delta \boldsymbol{x}_i))^2, \tag{13}$$

for $\Omega_{\mathcal{X}} = \cap_j \{\Delta \boldsymbol{x} : D_{\mathcal{H}_j}(\boldsymbol{x}, \boldsymbol{x} + \Delta \boldsymbol{x}) \leq \delta\} = \{\Delta \boldsymbol{x} : \max_j D_{\mathcal{H}_j}(\boldsymbol{x}, \boldsymbol{x} + \Delta \boldsymbol{x}) \leq \delta\}$.

**Reformulation and optimization.** Proposition 9 in the Appendix, allow for the reformulation of (12) as the following minimization problem

$$\min_{\substack{\boldsymbol{f}_i \in \mathcal{H}_i, \\ i=1,\ldots,D}} \frac{1}{n} \sum_{i=1}^{n} \left( \left| y - \sum_{j=1}^{D} \boldsymbol{f}_j(\boldsymbol{x}) \right| + \delta \sum_{j=1}^{D} \|\boldsymbol{f}_j\|_{\mathcal{H}_j} \right)^2,$$

which by inspection is very similar to the multiple kernel learning problem. A similar algorithm to Algorithm 1 also applies here and is described in Appendix D.

# 7 Related work

Adversarial attacks (Bruna et al., 2014; Goodfellow et al., 2015) can significantly degrade the performance of state-of-the-art models and adversarial training has emerged as one of the most effective strategies to prevent vulnerabilities.

**Adversarial training in neural networks.** Adversarial training is often studied in the context of neural networks and deep learning. Traditional methods for solving adversarial training require solving a min-max problem: minimizing a parameter vector while maximizing the error with a limited attack size. Examples of traditional methods to solve the inner maximization include the Fast Gradient Size Method (Goodfellow et al., 2015), and PGD-type of attacks (Madry et al., 2018). The problem can be solved by backpropagating through the inner loop (Madry et al., 2018). Recent efforts also aim to make adversarial training more efficient by approximate but faster update strategies. Indeed, recent works have sought to simplify adversarial training by relying on single-step (Wong et al., 2020) or latent-space perturbations (Park & Lee, 2021), rather than more costly multi-step procedures. Methods such as TRADES (Zhang et al., 2019) explore variants of robust optimization that trade off accuracy and robustness using tractable approximations. A recent (and more theoretically focused) study by Mousavi-Hosseini et al. (2025) proposes a method for learning multi-index models in an adversarially robust manner using two-layer neural networks—where the first layer can be trained in a standard way, and the second layer minimizes the adversarial loss.

We develop a framework not intended for neural networks, but one that we hope can inspire new methods in that domain. Our approach builds on theoretical insights from linear models, extending them to nonparametric settings.

**Adversarial training in linear models.** Adversarial training in linear models has been extensively studied over the past five years. The linear setting serves as a tractable framework for analyzing the behavior of deep neural networks. For instance, it has been used to study the robustness–accuracy trade-off (Ilyas et al., 2019; Tsipras et al., 2019) and the impact of overparameterization on adversarial robustness (Ribeiro & Schön, 2023). Asymptotic analyses have further examined adversarial training in binary classification and regression (Javanmard & Soltanolkotabi, 2022; Javanmard et al., 2020; Taheri et al., 2022), and also for random feature models (Hassani & Javanmard, 2024). Related work has also investigated Gaussian classification (Dan et al., 2020; Dobriban et al., 2023), the effect of dataset size on adversarial vulnerability (Min et al., 2021), and the behavior of $\ell_\infty$-attacks on linear classifiers (Yin et al., 2019). In this work, *we explore how these insights can be extended to function spaces and non-parametric models*. This allow for leveragin results from linear models, In particular, connections between $\ell_p$-norm adversarial training and ridge regresion, have been well established (Ribeiro et al., 2023). The optimization approach presented in Section 4 extends the method of Ribeiro et al. (2025) to nonparametric settings, while the generalization bounds derived in Section 5 adapt proof techniques from Ribeiro et al. (2023) and Xie and Huo (2024) to the context of nonparametric regression.

**Robust kernel regression.** There is a substantial body of work on robust kernel ridge regression, primarily aimed at improving robustness to outliers. These methods often rely on M-estimators (Debruyne et al., 2010; Hwang et al., 2015; Wibowo, 2009) or quantile regression (Li et al., 2007), and are typically optimized using iterative reweighted ridge regression schemes. While our method shares some algorithmic similarities, its purpose and formulation differ fundamentally. Our objective is not robustness to outliers, but robustness to adversarial perturbations. Moreover, *our formulation is not based on an M-estimator: the adversarial relaxation leads to an objective where both the loss and the regularization are jointly shaped by the perturbation set*, as shown in Equation (2). Another related line of work replaces the ridge penalty in kernel ridge regression with an $\ell_1$ penalty to promote sparsity in the observations (Allerbo, 2025; Feng et al., 2016; Roth, 2004). Alternatively, some approaches use max-norm (i.e., $\ell_\infty$) regularization to train kernelized support vector machines that are robust to label attacks (Russu et al., 2016). Our setting differs from both: we do not aim for sparsity, nor do we focus on robustness to label noise. Instead, we are concerned with robustness to structured perturbations in the input space.

## 8 Numerical experiments

The numerical experiments aim to evaluate the performance both in **clean** and **adversarially perturbed data**. We emphasize the optimal configuration for clean data does not necessarily coincide with that for adversarial robustness. For the clean data evaluation, following the intuition from our generalization analysis, we suggest decrease the perturbation radius $\delta$ as the sample size increases, $\delta \propto 1/\sqrt{n}$. In contrast, for robust evaluation, it is often beneficial to keep the adversarial radius fixed $\delta \propto \delta^{\text{test}}$.

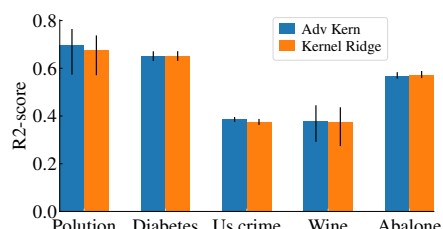

Figure 2: Kernel ridge *vs* adversarial kernel training. Compared using $R^2$ (higher is better).

**Evaluation on clean data: adaptativity to smoothness and noise levels.** We evaluate the proposed methods on both smooth and non-smooth target functions (same setting as in Figure 1). In particular, we investigate whether adversarial kernel training can adapt to the target's smoothness. Kernel ridge regression with cross-validation can automatically achieve such adaptation (Bach, 2021). Here, we perform a similar experiment, applying adversarial training with the default parameter choice ($\delta \propto n^{-1/2}$), and observe comparable behavior. The results in Figure 3 show that adversarial kernel training naturally adjusts to the regularity of the target function across different kernels. In these plots, the dashed line represents the linear approximation, whose slope indicates the observed efficiency rate (reported in Table S.1). Finally, Figure S.2 examines robustness to varying noise levels, comparing sine, square, and linear targets.

**Evaluation on clean data: performance on bechmarks.** In Figure 2, we compare the performance of adversarial kernel training with kernel ridge regression across several ***regression***

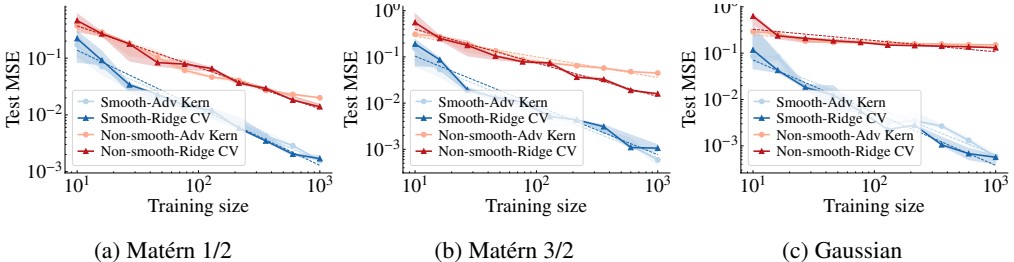

| | | (a) Matérn 1/2 | | (b) Matérn 3/2 | | (c) Gaussian |

Figure 3: We compare adversarial kernel training with cross-validated kernel ridge regression (Ridge CV). We show the test mean square error (MSE) vs training size $n$, for different kernels.

Table 2: Comparing methods under adversarial attacks (Abalone). Reported values are test set $R^2$ scores (higher is better), with bootstrapped interquartile ranges (Q1–Q3) shown in parentheses. The best result for each setting is highlighted in **boldface**.

| training method | no attack | test-time attack $\ell_2$ $\{\|\Delta \boldsymbol{x}\|_2 \le 0.01\}$ | $\{\|\Delta \boldsymbol{x}\|_2 \le 0.1\}$ | test-time attack $\ell_\infty$ $\{\|\Delta \boldsymbol{x}\|_\infty \le 0.01\}$ | $\{\|\Delta \boldsymbol{x}\|_\infty \le 0.1\}$ |
|---|---|---|---|---|---|
| **Adv Kern** ($\delta \propto n^{-1/2}$) | **0.57** (0.56–0.58) | **0.55** (0.54–0.57) | 0.39 (0.37–0.40) | **0.54** (0.52–0.55) | 0.13 (0.11–0.16) |
| Ridge Kernel ($\lambda$ cross-validated) | **0.57** (0.56–0.59) | **0.55** (0.53–0.56) | 0.26 (0.24–0.28) | 0.52 (0.51–0.54) | -0.16 (-0.20—0.13) |
| **Adv Kern** $\{\|\boldsymbol{d}\|_{\mathcal{H}} \le 0.01\}$ | 0.56 (0.55–0.58) | **0.55** (0.53–0.56) | **0.40** (0.39–0.42) | 0.53 (0.52–0.55) | 0.18 (0.16–0.20) |
| **Adv Kern** $\{\|\boldsymbol{d}\|_{\mathcal{H}} \le 0.1\}$ | 0.38 (0.37–0.39) | 0.38 (0.37–0.39) | 0.35 (0.34–0.36) | 0.37 (0.36–0.38) | **0.30** (0.29–0.31) |
| Adv Input $\{\|\Delta \boldsymbol{x}\|_2 \le 0.1\}$ | **0.57** (0.56–0.59) | **0.55** (0.53–0.56) | 0.27 (0.25–0.29) | 0.52 (0.51–0.54) | -0.14 (-0.17—0.11) |
| Adv Input $\{\|\Delta \boldsymbol{x}\|_\infty \le 0.1\}$ | **0.57** (0.55–0.58) | 0.54 (0.53–0.56) | 0.27 (0.25–0.29) | 0.52 (0.51–0.53) | -0.11 (-0.14—0.08) |

datasets. For kernel ridge regression, hyperparameters are selected via cross-validation over $\gamma \in \{10, 1, 0.1, 10^{-2}, 10^{-3}\}$ and $\lambda \in \{1, 0.1, 10^{-2}, 10^{-3}\}$. For adversarial kernel training, we use a default adversarial radius and select $\gamma$ from the same range using cross-validation. Thanks to its reduced hyperparameter space, *adversarial kernel training consistently performs on par with or better than kernel ridge regression across the evaluated benchmarks.*

**Robustness to adversarial attacks.** In Table 2, we focus on one of the benchmark datasets. In addition to the clean-data evaluation, we include four new training variants based on **feature-space adversarial kernel training** with fixed values of $\delta$—which is a natural choice when the objective is adversarial robustness. For comparison, we also include an **input-space adversarial training** baseline with $\delta^{\text{train}} = 0.05$, following Equation (1), trained for 300 epochs using Adam. All methods are evaluated both on clean data and under test-time adversarial attacks in the $\ell_2$ and $\ell_\infty$ norms. Results on additional datasets are provided in the Appendix.

Notably, although our proposed method is trained with perturbations in feature space, *it remains highly robust to input-space attacks, outperforming approaches explicitly trained for input robustness.* We hypothesize that this arises from the feature-perturbed formulation offering more favorable optimization properties. Furthermore, comparing models feture-perturbed adversarially trained models with $\|\boldsymbol{d}\|_{\mathcal{H}} \le 0.01$ and $\|\boldsymbol{d}\|_{\mathcal{H}} \le 0.1$ illustrates the trade-off between clean and adversarial accuracy: as the feature-space adversarial radius $\delta$ increases, the model becomes more robust to input-space attacks—particularly when the attacker has a large budget, e.g., $\|\Delta \boldsymbol{x}\|_\infty \le 0.1$—albeit at the cost of a drop in clean performance. Interestingly, our method still surpasses direct input-space adversarial training even under the same test-time threat model.

## 9 Conclusion and future work

We covered several aspects of the proposed method. Our goal was to present a unified perspective that makes the method broadly applicable, by including optimization, generalization and robustness properties. Different aspects however deserve deeper investigation. Further directions include improving the optimization focusing large-scale systems. The proposed generalization bounds also deserve more detailed analysis and as future work, we aim to fully explore the results in the misspecified and the random design setting. Another promising direction is generalizing to classification and to further study the multiple kernel learning scenario. The ideas here also naturally extend to infinitely wide neural networks, which can often be analyzed using RKHS tools. A key example is the neural tangent kernel (NTK) framework (Jacot et al., 2018). Moreover, it would be interesting to explore adversarial training for infinitely wide networks as a tool to guide architecture and hyperparameter selection, following similar ideas to Yang et al. (2021).

## Acknowledgments and Disclosure of Funding

This work was partially developed during AHR visit to the SIERRA team at INRIA. We would like to thank Dominik Baumann, Oskar Allerbo and Elis Stefansson for giving feedback on the early versions of this manuscript. We would also like to thank Bruno Loureiro for the interesting discussions around the theme. TBS and DZ are financially supported by the Swedish Research Council, with the projects Deep probabilistic regression–new models and learning algorithms (project number 2021-04301) and Robust learning methods for out-of-distribution tasks (project number 2021-05022); and, by the Wallenberg AI, Autonomous Systems and Software Program (WASP) funded by Knut and Alice Wallenberg Foundation. TBS, AHR and DZ are partially funded by the Kjell och Märta Beijer Foundation. FB by the National Research Agency, under the France 2030 program with the reference "PR[AI]RIE-PSAI" (ANR-23-IACL-0008).

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

# Table of Contents

# A    Problem formulation

## A.1    Proof of Proposition 1

Let $\Delta \boldsymbol{x} \in \Omega_{\mathcal{X}}$ then by definition

$$\|\boldsymbol{\phi}(\boldsymbol{x}) - \boldsymbol{\phi}(\boldsymbol{x} + \Delta \boldsymbol{x})\|_{\mathcal{H}} = D_{\mathcal{H}}(\boldsymbol{x}, \boldsymbol{x} + \Delta \boldsymbol{x}) \leq \delta$$

hence $(\boldsymbol{\phi}(\boldsymbol{x}) - \boldsymbol{\phi}(\boldsymbol{x} + \Delta \boldsymbol{x})) \in \Omega_{\mathcal{H}}$ and it follows that:

$$\max_{\Delta \boldsymbol{x} \in \Omega_{\mathcal{X}}} (y_i - \boldsymbol{f}(\boldsymbol{x} + \Delta \boldsymbol{x}))^2 = \max_{\Delta \boldsymbol{x} \in \Omega_{\mathcal{X}}} (y_i - \langle \boldsymbol{f}, \boldsymbol{\phi}(\boldsymbol{x} + \Delta \boldsymbol{x}) \rangle)^2$$

$$= \max_{\substack{\boldsymbol{d} = \boldsymbol{\phi}(\boldsymbol{x} + \Delta \boldsymbol{x}) - \boldsymbol{\phi}(\boldsymbol{x}), \\ \Delta \boldsymbol{x} \in \Omega_{\mathcal{X}}}} (y_i - \langle \boldsymbol{f}, \boldsymbol{\phi}(\boldsymbol{x}) + \boldsymbol{d} \rangle)^2$$

$$\leq \max_{\boldsymbol{d} \in \Omega_{\mathcal{H}}} (y_i - \langle \boldsymbol{f}, \boldsymbol{\phi}(\boldsymbol{x}) + \boldsymbol{d} \rangle)^2$$

## A.2    Proof of regions in Table 1

**Linear kernel.**    For the linear kernel:

$$k(\boldsymbol{x}, \boldsymbol{x}') = \boldsymbol{x}^{\top} \boldsymbol{x}$$

we have the evaluation functional $\boldsymbol{\phi}(\boldsymbol{x}) = \boldsymbol{x}$, and the induced RKHS space $\mathcal{H} = \mathbb{R}^p$. Hence:

$$D_{\mathcal{H}}(\boldsymbol{x}, \boldsymbol{x} + \Delta \boldsymbol{x}) = \|\Delta \boldsymbol{x}\|_2$$

and $\Omega_{\mathcal{X}} = \{\Delta \boldsymbol{x} : \|\Delta \boldsymbol{x}\|_2 \leq \delta\}$.

**Translation invariant kernels.**    Now, let us have any kernel

$$k(\boldsymbol{x}, \boldsymbol{x}') = f(\|\boldsymbol{x} - \boldsymbol{x}'\|)$$

for which $f$ is an strictly decreasing function with $f(0) = 1$ we use that:

$$D_{\mathcal{H}}(\boldsymbol{x}, \boldsymbol{x}') = k(\boldsymbol{x}, \boldsymbol{x}) - 2k(\boldsymbol{x}, \boldsymbol{x}') + k(\boldsymbol{x}', \boldsymbol{x}') = 2 - 2f(\|\boldsymbol{x} - \boldsymbol{x}'\|)$$

where in the second inequality we plugged in the definition of kernel. Now, since $f$ monotonic, $D_{\mathcal{H}}(\boldsymbol{x}, \boldsymbol{x} + \Delta \boldsymbol{x}) \leq \delta$ implies that

$$\|\boldsymbol{x} - \boldsymbol{x}'\| \leq f^{-1}\left(1 - \frac{\delta}{2}\right)$$

Now we can apply this result to kernels of interest, particularly, Gaussian, Laplace and Matérn kernels.

**Gaussian kernel.**    The gaussian kernel

$$k(\boldsymbol{x}, \boldsymbol{x}') = \exp(-\gamma \|x - x'\|_2^2),$$

falls into the above scenario and, hence, the above derivation implies that

$$\exp(-\gamma \|x - x'\|_2^2) \geq 1 - \frac{\delta}{2}$$

$$\|x - x'\|_2 \leq \sqrt{\frac{1}{\gamma} \log \frac{1}{1 - \frac{\delta}{2}}}.$$

and the result follows. Very similar derivations hold for the Laplacian and exponential kernels.

**Polynomial.** If $\Delta \boldsymbol{x} = \boldsymbol{x}' - \boldsymbol{x} \in \Omega_{\mathcal{X}}$ then

$$\delta^2 \geq D_{\mathcal{H}}(\boldsymbol{x}, \boldsymbol{x}')$$
$$= k(\boldsymbol{x}, \boldsymbol{x}) + k(\boldsymbol{x}', \boldsymbol{x}') - 2k(\boldsymbol{x}, \boldsymbol{x}').$$

Now, rearranging and exponentiation by $1/d$,

$$2^{1/d} k(\boldsymbol{x}, \boldsymbol{x}')^{1/d} \geq (k(\boldsymbol{x}, \boldsymbol{x}) + k(\boldsymbol{x}', \boldsymbol{x}') - \delta^2)^{1/d}$$
$$\overset{(a)}{\geq} (k(\boldsymbol{x}, \boldsymbol{x}) + k(\boldsymbol{x}', \boldsymbol{x}'))^{1/d} - \delta^{2/d}$$
$$\overset{(b)}{\geq} 2^{1/d}(k(\boldsymbol{x}, \boldsymbol{x})k(\boldsymbol{x}', \boldsymbol{x}'))^{1/(2d)} - \delta^{2/d}.$$

Where, (a) follows from applying Proposition 4 with $z = (\frac{\delta^2}{k(\boldsymbol{x},\boldsymbol{x})+k(\boldsymbol{x}',\boldsymbol{x}')})^{1/d}$. And, (b) follows from using Proposition 5 with $\zeta = 1$. Therefore:

$$\delta^{2/d} 2^{-1/d} \geq -k(\boldsymbol{x}, \boldsymbol{x}')^{1/d} + (k(\boldsymbol{x}, \boldsymbol{x})k(\boldsymbol{x}', \boldsymbol{x}'))^{1/(2d)}$$
$$= \underbrace{\frac{1}{2}k(\boldsymbol{x}, \boldsymbol{x})^{1/d} + \frac{1}{2}k(\boldsymbol{x}', \boldsymbol{x}')^{1/d} - k(\boldsymbol{x}, \boldsymbol{x}')^{1/d}}_{A} - \underbrace{\frac{1}{2}(k(\boldsymbol{x}, \boldsymbol{x})^{1/(2d)} - k(\boldsymbol{x}', \boldsymbol{x}')^{1/(2d)})^2}_{B}$$

Where the last equality follows from summing and subtracting $\frac{1}{2}k(\boldsymbol{x}, \boldsymbol{x})^{1/d} + \frac{1}{2}k(\boldsymbol{x}', \boldsymbol{x}')^{1/d}$ and regrouping. Now, we will rewrite $A$ and $B$, using the definition of polinomial kernel, i.e. $k(\boldsymbol{x}, \boldsymbol{x}')^{1/d} = 1 + \boldsymbol{x}^\top \boldsymbol{x}'$. On one hand,

$$2A = k(\boldsymbol{x}, \boldsymbol{x})^{1/d} + k(\boldsymbol{x}', \boldsymbol{x}')^{1/d} - 2k(\boldsymbol{x}, \boldsymbol{x}')^{1/d} = \|\Delta \boldsymbol{x}\|^2,$$

and, on the other hand,

$$2B = (\sqrt{1 + \|\boldsymbol{x}\|^2} - \sqrt{1 + \|\boldsymbol{x}'\|^2})^2$$
$$= \frac{(\|\boldsymbol{x}\|^2 - \|\boldsymbol{x}'\|^2)^2}{(\sqrt{1 + \|\boldsymbol{x}\|^2} - \sqrt{1 + \|\boldsymbol{x}'\|^2})^2}$$
$$= \frac{(\|\boldsymbol{x}\| - \|\boldsymbol{x}'\|)^2 (\|\boldsymbol{x}\| + \|\boldsymbol{x}'\|)^2}{(\sqrt{1 + \|\boldsymbol{x}\|^2} + \sqrt{1 + \|\boldsymbol{x}'\|^2})^2}$$
$$\leq \frac{\|\Delta \boldsymbol{x}\|^2 (\|\boldsymbol{x}\| + \|\boldsymbol{x}'\|)^2}{(\sqrt{1 + \|\boldsymbol{x}\|^2} + \sqrt{1 + \|\boldsymbol{x}'\|^2})^2}$$
$$\leq \frac{\|\Delta \boldsymbol{x}\|^2 (\|\boldsymbol{x}\| + \|\boldsymbol{x}'\|)^2}{2 + \|\boldsymbol{x}\|^2 + \|\boldsymbol{x}'\|^2}$$
$$\leq \frac{2\|\Delta \boldsymbol{x}\|^2 (\|\boldsymbol{x}\|^2 + \|\boldsymbol{x}'\|^2)}{2 + \|\boldsymbol{x}\|^2 + \|\boldsymbol{x}'\|^2}$$

Hence,

$$\delta^{2/d} 2^{-1/d} \geq A - B$$
$$\delta^{2/d} 2^{-1/d} \geq \|\Delta \boldsymbol{x}\|^2 \frac{1}{2 + \|\boldsymbol{x}\|^2 + \|\boldsymbol{x}'\|^2}$$
$$\delta^{2/d} 2^{-1/d}(2 + \|\boldsymbol{x}\|^2 + \|\boldsymbol{x}'\|^2) \geq \|\Delta \boldsymbol{x}\|^2$$
$$\delta^{1/d} 2^{-1/(2d)} \sqrt{2 + \|\boldsymbol{x}\|^2 + \|\boldsymbol{x}'\|^2} \geq \|\Delta \boldsymbol{x}\|$$

Putting everything together, we obtain

$$C \delta^{1/d} \geq \|\Delta \boldsymbol{x}\|$$

for $C = 2^{-1/(2d)} \sqrt{2 + \|\boldsymbol{x}\|^2 + \|\boldsymbol{x}'\|^2}$. Next, we prove the proposition that was used.

**Proposition 4.** For any $0 < z < 1$ and $d$ any positive integer, we have that

$$(1-z)^d \leq 1 - z^d.$$

*Proof.* We can prove it recursively. The result is trivial for $d = 1$, now if assume that:

$$(1-z)^{d-1} \leq 1 - z^{d-1},$$

since $(1-z) > 0$ we have

$$(1-z)^d \leq (1-z^{d-1})(1-z) \leq 1 - z - z^{d-1} + z^d \leq 1 + z^d$$

The result follows. $\qquad\square$

## B    Convergence rates

### B.1    Proof of Theorem 2

In our developments, we will also use the following proposition.

**Proposition 5.** For any $\zeta > 0$,

$$ab \leq \frac{1}{2\zeta}a^2 + \frac{\zeta}{2}b^2$$

For kernel ridge regression we are minimizing the cost function:

$$L(\boldsymbol{f}) = \frac{1}{n}\sum_{i=1}^{n}(y_i - \boldsymbol{f}(\boldsymbol{x}_i))^2 + \lambda\|\boldsymbol{f}\|_{\mathcal{H}}^2$$

using that $y_i = \boldsymbol{f}^*(x_i) + \sigma w_i$ we obtain for an estimated function $\widehat{\boldsymbol{f}}$ that:

$$L(\widehat{\boldsymbol{f}}) = \frac{1}{n}\sum_{i}(\boldsymbol{f}^*(x_i) + \sigma w_i - \widehat{\boldsymbol{f}}(x_i))^2 + \lambda\|\widehat{\boldsymbol{f}}\|_{\mathcal{H}}^2$$

$$\overset{(a)}{=} \mathcal{E}(\widehat{\boldsymbol{f}} - \boldsymbol{f}^*) - \frac{2\sigma}{n}\sum_{i=1}^{n}w_i(\widehat{\boldsymbol{f}}(\boldsymbol{x}_i) - \boldsymbol{f}^*(\boldsymbol{x}_i)) + \sigma^2 + \lambda\|\widehat{\boldsymbol{f}}\|_{\mathcal{H}}^2$$

where in (a), we used that $\frac{1}{n}\sum_{i=1}^{n}w_i^2 = 1$ and $\mathcal{E}(\widehat{\boldsymbol{f}} - \boldsymbol{f}^*) = \frac{1}{n}\sum_{i=1}^{n}(\widehat{\boldsymbol{f}}(\boldsymbol{x}_i) - \boldsymbol{f}^*(\boldsymbol{x}_i))^2$. By definition $L(\widehat{\boldsymbol{f}}) < L(\boldsymbol{f}^*)$ and by a similar procedure we can obtain that $L(\boldsymbol{f}^*) = \sigma^2 + \lambda\|\boldsymbol{f}^*\|_{\mathcal{H}}^2$. Hence, simple algebraic manipulation shows that:

$$\mathcal{E}(\widehat{\boldsymbol{f}} - \boldsymbol{f}^*) \leq \frac{2\sigma}{n}\sum_{i=1}^{n}w_i(\widehat{\boldsymbol{f}}(\boldsymbol{x}_i) - \boldsymbol{f}^*(\boldsymbol{x}_i)) + \lambda\left(\|\boldsymbol{f}^*\|_{\mathcal{H}}^2 - \|\widehat{\boldsymbol{f}}\|_{\mathcal{H}}^2\right) \qquad\text{(S.1)}$$

**Bounds as a function of $\gamma$.**    By the definition of $\gamma$ in the theorem assumption:

$$\frac{1}{n}\sum_{i=1}^{n}w_i(\widehat{\boldsymbol{f}}(\boldsymbol{x}_i) - \boldsymbol{f}^*(\boldsymbol{x}_i)) \leq \gamma\|\widehat{\boldsymbol{f}} - \boldsymbol{f}^*\|_{\mathcal{H}}.$$

Hence:

$$0 \leq \mathcal{E}(\widehat{\boldsymbol{f}} - \boldsymbol{f}^*) \leq 2\sigma\gamma\|\widehat{\boldsymbol{f}} - \boldsymbol{f}^*\|_{\mathcal{H}} + \lambda(\|\boldsymbol{f}^*\|_{\mathcal{H}}^2 - \|\widehat{\boldsymbol{f}}\|_{\mathcal{H}}^2) \qquad\text{(S.2a)}$$

$$\leq \frac{2\gamma\sigma}{\|\boldsymbol{f}^*\|_{\mathcal{H}}}\|\boldsymbol{f}^*\|_{\mathcal{H}}\|\widehat{\boldsymbol{f}} - \boldsymbol{f}^*\|_{\mathcal{H}} + \lambda(\|\boldsymbol{f}^*\|_{\mathcal{H}} - \|\widehat{\boldsymbol{f}}\|_{\mathcal{H}})(\|\boldsymbol{f}^*\|_{\mathcal{H}} + \|\widehat{\boldsymbol{f}}\|_{\mathcal{H}}) \qquad\text{(S.2b)}$$

$$\leq (\|\boldsymbol{f}^*\|_{\mathcal{H}} + \|\widehat{\boldsymbol{f}}\|_{\mathcal{H}})\left[\left(\frac{2\gamma\sigma}{\|\boldsymbol{f}^*\|_{\mathcal{H}}} + \lambda\right)\|\boldsymbol{f}^*\|_{\mathcal{H}} - \lambda\|\widehat{\boldsymbol{f}}\|_{\mathcal{H}}\right] \qquad\text{(S.2c)}$$

And therefore:

$$\|\widehat{\boldsymbol{f}}\|_{\mathcal{H}} \leq \left(1 + \frac{\gamma\sigma}{\lambda\|\boldsymbol{f}^*\|_{\mathcal{H}}}\right)\|\boldsymbol{f}^*\|_{\mathcal{H}}$$

Replacing this in Equation (S.2c)

$$\mathcal{E}(\widehat{\boldsymbol{f}} - \boldsymbol{f}^*) \le \left(2 + \frac{\gamma\sigma}{\lambda\|\boldsymbol{f}^*\|_{\mathcal{H}}}\right)\|\boldsymbol{f}^*\|_{\mathcal{H}}\left[\left(\frac{2\gamma\sigma}{\|\boldsymbol{f}^*\|_{\mathcal{H}}} + \lambda\right)\|\boldsymbol{f}^*\|_{\mathcal{H}} - \lambda\|\widehat{\boldsymbol{f}}\|_{\mathcal{H}}\right]$$

$$\le \left(2 + \frac{\gamma\sigma}{\lambda\|\boldsymbol{f}^*\|_{\mathcal{H}}}\right)\|\boldsymbol{f}^*\|_{\mathcal{H}}\left(\frac{2\gamma\sigma}{\|\boldsymbol{f}^*\|_{\mathcal{H}}} + \lambda\right)\|\boldsymbol{f}^*\|_{\mathcal{H}}$$

$$\le 2\lambda\left(\|\boldsymbol{f}^*\|_{\mathcal{H}} + \frac{\gamma\sigma}{\lambda}\right)^2$$

And the result follows.

**Bounds as a function of $\beta$.** By the definition of $\beta$ in the theorem assumption:

$$\frac{1}{n}\sum_{i=1}^{n}w_i(\widehat{\boldsymbol{f}}(\boldsymbol{x}_i) - \boldsymbol{f}^*(\boldsymbol{x}_i)) \le \beta\sqrt{\mathcal{E}(\widehat{\boldsymbol{f}} - \boldsymbol{f}^*)}.$$

Hence:

$$\mathcal{E}(\widehat{\boldsymbol{f}} - \boldsymbol{f}^*) \le 2\sigma\beta\sqrt{\mathcal{E}(\widehat{\boldsymbol{f}} - \boldsymbol{f}^*)} + \lambda(\|\boldsymbol{f}^*\|_{\mathcal{H}}^2 - \|\widehat{\boldsymbol{f}}\|_{\mathcal{H}}^2)$$

$$\overset{(a)}{\le} \sigma^2\beta^2 + \frac{1}{2}\mathcal{E}(\widehat{\boldsymbol{f}} - \boldsymbol{f}^*) + \lambda\|\boldsymbol{f}^*\|_{\mathcal{H}}^2$$

where (a) follows from Proposition 5 with $\zeta = 1$. Hence, using $R = \|\boldsymbol{f}^*\|_{\mathcal{H}}$ we obtain:

$$\mathcal{E}(\widehat{\boldsymbol{f}} - \boldsymbol{f}^*) \le 2\sigma^2\beta^2 + 2\lambda R^2$$

## B.2 Proof of Theorem 1

For adversarial kernel regression we are minimizing the cost function:

$$L(\boldsymbol{f}) = \frac{1}{n}(|y_i - \boldsymbol{f}(\boldsymbol{x}_i)| + \delta\|\boldsymbol{f}\|_{\mathcal{H}})^2,$$

$$= \frac{1}{n}\sum_{i=1}^{n}(y_i - \boldsymbol{f}(\boldsymbol{x}_i))^2 + \frac{2\delta}{n}\|\boldsymbol{f}\|_{\mathcal{H}}\sum_{i=1}^{n}|y_i - \boldsymbol{f}(\boldsymbol{x}_i)| + \delta^2\|\boldsymbol{f}\|_{\mathcal{H}}^2.$$

Similarly to the proof of Theorem 2 using that $y_i = \boldsymbol{f}^*(x_i) + \sigma w_i$ we obtain for an estimated function $\widehat{\boldsymbol{f}}$ that:

$$L(\widehat{\boldsymbol{f}}) = \mathcal{E}(\widehat{\boldsymbol{f}} - \boldsymbol{f}^*) - \frac{2\sigma}{n}\sum_{i=1}^{n}w_i(\widehat{\boldsymbol{f}}(\boldsymbol{x}_i) - \boldsymbol{f}^*(\boldsymbol{x}_i)) + \sigma^2 + \frac{2\delta}{n}\|\widehat{\boldsymbol{f}}\|_{\mathcal{H}}\sum_{i=1}^{n}|\boldsymbol{f}^*(\boldsymbol{x}_i) + \sigma w_i - \widehat{\boldsymbol{f}}(\boldsymbol{x}_i)| + \delta^2\|\widehat{\boldsymbol{f}}\|_{\mathcal{H}}^2$$

similarly,

$$L(\boldsymbol{f}^*) = \sigma^2 + 2\delta\sigma\|\boldsymbol{f}^*\|_{\mathcal{H}}\left(\frac{1}{n}\sum_i|w_i|\right) + \delta^2\|\boldsymbol{f}^*\|_{\mathcal{H}}^2$$

By definition $L(\widehat{\boldsymbol{f}}) < L(\boldsymbol{f}^*)$. Hence, simple algebraic manipulation shows that:

$$\mathcal{E}(\widehat{\boldsymbol{f}} - \boldsymbol{f}^*) \le \frac{2\sigma}{n}\sum_{i=1}^{n}w_i(\widehat{\boldsymbol{f}}(\boldsymbol{x}_i) - \boldsymbol{f}^*(\boldsymbol{x}_i)) + \frac{2\delta}{n}\|\widehat{\boldsymbol{f}}\|_{\mathcal{H}}\sum_i(|\sigma w_i| - |\boldsymbol{f}^*(\boldsymbol{x}_i) + \sigma w_i - \widehat{\boldsymbol{f}}(\boldsymbol{x}_i)|)$$

$$+ 2\delta\sigma(\|\boldsymbol{f}^*\|_{\mathcal{H}} - \|\widehat{\boldsymbol{f}}\|_{\mathcal{H}})\left(\frac{1}{n}\sum_i|w_i|\right) + \delta^2\left(\|\boldsymbol{f}^*\|_{\mathcal{H}}^2 - \|\widehat{\boldsymbol{f}}\|_{\mathcal{H}}^2\right).$$

Now using the relation between norms $\|\boldsymbol{z}\|_1 \le \sqrt{\dim(\boldsymbol{z})}\|\boldsymbol{z}\|_2$ we have that:

$$\frac{1}{n}\sum_i(|\sigma w_i| - |\boldsymbol{f}^*(\boldsymbol{x}_i) + \sigma w_i - \widehat{\boldsymbol{f}}(\boldsymbol{x}_i)|) \le \frac{1}{n}\sum_i|\boldsymbol{f}^*(\boldsymbol{x}_i) - \widehat{\boldsymbol{f}}(\boldsymbol{x}_i)| \le \sqrt{\mathcal{E}(\widehat{\boldsymbol{f}} - \boldsymbol{f}^*)}$$

$$\frac{1}{n}\sum_i|w_i| \le \frac{1}{\sqrt{n}}\sum_{i=1}^{n}w_i^2 = 1,$$

and it follows that:

$$\mathcal{E}(\widehat{\boldsymbol{f}}-\boldsymbol{f}^*) \leq \frac{2\sigma}{n}\sum_{i=1}^n w_i(\widehat{\boldsymbol{f}}(\boldsymbol{x}_i)-\boldsymbol{f}^*(\boldsymbol{x}_i))+2\delta\|\widehat{\boldsymbol{f}}\|_{\mathcal{H}}\sqrt{\mathcal{E}(\widehat{\boldsymbol{f}}-\boldsymbol{f}^*)}+2\delta\sigma(\|\boldsymbol{f}^*\|_{\mathcal{H}}-\|\widehat{\boldsymbol{f}}\|_{\mathcal{H}})+\delta^2\left(\|\boldsymbol{f}^*\|_{\mathcal{H}}^2-\|\widehat{\boldsymbol{f}}\|_{\mathcal{H}}^2\right)$$

$$\text{(S.3)}$$

**Bounds as a function of $\gamma$.** Now, by the definition of $\gamma$ in the theorem assumption:

$$\frac{1}{n}\sum_{i=1}^n w_i(\widehat{\boldsymbol{f}}(\boldsymbol{x}_i)-\boldsymbol{f}^*(\boldsymbol{x}_i)) \leq \gamma\|\widehat{\boldsymbol{f}}-\boldsymbol{f}^*\|_{\mathcal{H}}.$$

Hence:

$$\mathcal{E}(\widehat{\boldsymbol{f}}-\boldsymbol{f}^*) \leq 2\sigma\gamma\|\boldsymbol{f}^*-\widehat{\boldsymbol{f}}\|+2\delta\|\widehat{\boldsymbol{f}}\|_{\mathcal{H}}\sqrt{\mathcal{E}(\widehat{\boldsymbol{f}}-\boldsymbol{f}^*)}+2\delta\sigma(\|\boldsymbol{f}^*\|_{\mathcal{H}}-\|\widehat{\boldsymbol{f}}\|_{\mathcal{H}})+\delta^2\left(\|\boldsymbol{f}^*\|_{\mathcal{H}}^2-\|\widehat{\boldsymbol{f}}\|_{\mathcal{H}}^2\right).$$

Now usign Proposition 5 with $\zeta=1$ and rearranging, we can obtain a bound on $\mathcal{E}$:

$$\begin{aligned}
\mathcal{E}(\widehat{\boldsymbol{f}}-\boldsymbol{f}^*) &\leq 4\sigma\gamma\|\boldsymbol{f}^*-\widehat{\boldsymbol{f}}\|_{\mathcal{H}}+4\delta^2\|\widehat{\boldsymbol{f}}\|_{\mathcal{H}}^2+4\delta\sigma(\|\boldsymbol{f}^*\|_{\mathcal{H}}-\|\widehat{\boldsymbol{f}}\|_{\mathcal{H}})+2\delta^2\left(\|\boldsymbol{f}^*\|_{\mathcal{H}}^2-\|\widehat{\boldsymbol{f}}\|_{\mathcal{H}}^2\right)\\
&\leq 4\sigma\gamma\|\boldsymbol{f}^*-\widehat{\boldsymbol{f}}\|_{\mathcal{H}}+4\delta\sigma(\|\boldsymbol{f}^*\|_{\mathcal{H}}-\|\widehat{\boldsymbol{f}}\|_{\mathcal{H}})+2\delta^2\left(\|\boldsymbol{f}^*\|_{\mathcal{H}}^2+\|\widehat{\boldsymbol{f}}\|_{\mathcal{H}}^2\right)\\
&\leq 4\sigma\gamma\|\boldsymbol{f}^*-\widehat{\boldsymbol{f}}\|_{\mathcal{H}}+4\delta\sigma\|\boldsymbol{f}^*-\widehat{\boldsymbol{f}}\|_{\mathcal{H}}+2\delta^2\left(\|\boldsymbol{f}^*\|_{\mathcal{H}}^2+\|\widehat{\boldsymbol{f}}\|_{\mathcal{H}}^2\right)\\
&\leq 4\sigma(\gamma+\delta)\|\boldsymbol{f}^*-\widehat{\boldsymbol{f}}\|_{\mathcal{H}}+2\delta^2\left(\|\boldsymbol{f}^*\|_{\mathcal{H}}^2+\|\widehat{\boldsymbol{f}}\|_{\mathcal{H}}^2\right)
\end{aligned}$$

Now, we use a result that we prove later in Proposition 6, Eq. (S.5), particularly, we have that $\|\widehat{\boldsymbol{f}}\|_{\mathcal{H}} \leq (1+\frac{\gamma}{\delta})\|\boldsymbol{f}^*\|_{\mathcal{H}}$. In this case,

$$\mathcal{E}(\widehat{\boldsymbol{f}}-\boldsymbol{f}^*) \leq 8\sigma\delta(1+\gamma/\delta)^2\|\boldsymbol{f}^*\|_{\mathcal{H}}+4\delta^2(1+\gamma/\delta)^2\|\boldsymbol{f}^*\|_{\mathcal{H}}^2$$

Using that $\|\boldsymbol{f}^*\|_{\mathcal{H}}=R$ we obtain

$$\mathcal{E}(\widehat{\boldsymbol{f}}-\boldsymbol{f}^*) \leq 4\delta R(2\sigma+\delta R)(1+\gamma/\delta)^2$$

**Bounds as a function of $\beta$.** Alternatively, by the definition of $\beta$ in the theorem assumption:

$$\frac{1}{n}\sum_{i=1}^n w_i(\widehat{\boldsymbol{f}}(\boldsymbol{x}_i)-\boldsymbol{f}^*(\boldsymbol{x}_i)) \leq \beta\sqrt{\mathcal{E}(\widehat{\boldsymbol{f}}-\boldsymbol{f}^*)}.$$

Hence, from (S.3) we obtain

$$\begin{aligned}
\mathcal{E}(\widehat{\boldsymbol{f}}-\boldsymbol{f}^*) &\leq 2\sigma\beta\sqrt{\mathcal{E}(\widehat{\boldsymbol{f}}-\boldsymbol{f}^*)}+2\delta\|\widehat{\boldsymbol{f}}\|_{\mathcal{H}}\sqrt{\mathcal{E}(\widehat{\boldsymbol{f}}-\boldsymbol{f}^*)}+2\delta\sigma(\|\boldsymbol{f}^*\|_{\mathcal{H}}-\|\widehat{\boldsymbol{f}}\|_{\mathcal{H}})+\delta^2\left(\|\boldsymbol{f}^*\|_{\mathcal{H}}^2-\|\widehat{\boldsymbol{f}}\|_{\mathcal{H}}^2\right)\\
&\leq 2\sigma\beta\sqrt{\mathcal{E}(\widehat{\boldsymbol{f}}-\boldsymbol{f}^*)}+2\delta\|\widehat{\boldsymbol{f}}\|_{\mathcal{H}}\sqrt{\mathcal{E}(\widehat{\boldsymbol{f}}-\boldsymbol{f}^*)}+2\delta\sigma\|\boldsymbol{f}^*\|_{\mathcal{H}}+\delta^2\|\boldsymbol{f}^*\|_{\mathcal{H}}^2\\
&\leq 4\sigma\beta\sqrt{\mathcal{E}(\widehat{\boldsymbol{f}}-\boldsymbol{f}^*)}+2\delta\|\boldsymbol{f}^*\|_{\mathcal{H}}\sqrt{\mathcal{E}(\widehat{\boldsymbol{f}}-\boldsymbol{f}^*)}+2\delta\sigma\|\boldsymbol{f}^*\|_{\mathcal{H}}+\delta^2\|\boldsymbol{f}^*\|_{\mathcal{H}}^2
\end{aligned}$$

Where in the last equation we used Proposition 6, Eq. (S.6): $\delta\|\widehat{\boldsymbol{f}}\|_{\mathcal{H}} \leq \delta\|\boldsymbol{f}^*\|_{\mathcal{H}}+\beta\sigma$. Now, applying Proposition 5 with $\zeta=2$ for each of the first two terms:

$$\mathcal{E}(\widehat{\boldsymbol{f}}-\boldsymbol{f}^*) \leq 8\sigma^2\beta^2+\frac{1}{4}\mathcal{E}(\widehat{\boldsymbol{f}}-\boldsymbol{f}^*)+4\delta^2\|\boldsymbol{f}^*\|_{\mathcal{H}}^2+\frac{1}{4}\mathcal{E}(\widehat{\boldsymbol{f}}-\boldsymbol{f}^*)+2\delta\sigma\|\boldsymbol{f}^*\|_{\mathcal{H}}+\delta^2\|\boldsymbol{f}^*\|_{\mathcal{H}}^2$$

And therefore:

$$\mathcal{E}(\widehat{\boldsymbol{f}}-\boldsymbol{f}^*) \leq 16\sigma^2\beta^2+4\sigma\delta R+10\delta^2 R^2 \tag{S.4}$$

Next, we prove the relation between the norms $\|\widehat{\boldsymbol{f}}\|_{\mathcal{H}}$ and $\|\boldsymbol{f}^*\|_{\mathcal{H}}$ that were used in the proof above.

> **Proposition 6.** If the conditions of Theorem 1 are satisfied. On the one hand:
>
> $$\delta\|\widehat{\boldsymbol{f}}\|_{\mathcal{H}} \leq (\delta + \gamma)\|\boldsymbol{f}^*\|_{\mathcal{H}} \tag{S.5}$$
>
> on the other hand:
>
> $$\delta\|\widehat{\boldsymbol{f}}\|_{\mathcal{H}} \leq \delta\|\boldsymbol{f}^*\|_{\mathcal{H}} + \sigma\beta \tag{S.6}$$

*Proof.* For the adversarial kernel regression loss:

$$L(\boldsymbol{f}) = \frac{1}{n}\sum_{i=1}^{n}(|y_i - \boldsymbol{f}(\boldsymbol{x}_i)| + \delta\|\boldsymbol{f}\|_{\mathcal{H}})^2,$$

$$= \frac{1}{n}\sum_{i}(y_i - \langle\boldsymbol{f}, \boldsymbol{\phi}(\boldsymbol{x}_i)\rangle)^2 + \frac{2\delta}{n}\|\boldsymbol{f}\|_{\mathcal{H}}\sum_{i}|y_i - \langle\boldsymbol{f}, \boldsymbol{\phi}(\boldsymbol{x}_i)\rangle| + \delta^2\|\boldsymbol{f}\|_{\mathcal{H}}^2$$

the estimator $\widehat{\boldsymbol{f}}$ minimizes $L(\boldsymbol{f})$, and satisfies the subderivative optimality condition $\partial L(\widehat{\boldsymbol{f}}) = 0$. We have

$$0 = \frac{1}{n}\sum_{i}(\langle\widehat{\boldsymbol{f}}, \boldsymbol{\phi}(\boldsymbol{x}_i)\rangle - y_i)\boldsymbol{\phi}(\boldsymbol{x}_i) + \frac{\delta\|\widehat{\boldsymbol{f}}\|_{\mathcal{H}}}{n}\frac{1}{n}\sum_{i=1}^{n}\widehat{z}_i\boldsymbol{\phi}(\boldsymbol{x}_i) + \delta\left(\frac{1}{n}\sum_{i}|y_i - \langle\widehat{\boldsymbol{f}}, \boldsymbol{\phi}(\boldsymbol{x}_i)\rangle| + \delta\|\widehat{\boldsymbol{f}}\|_{\mathcal{H}}\right)\widehat{\boldsymbol{w}},$$

where $\widehat{z}_i \in \partial|\langle\widehat{\boldsymbol{f}}, \boldsymbol{\phi}(\boldsymbol{x}_i)\rangle - y_i|$ and $\widehat{\boldsymbol{w}} \in \partial\|\widehat{\boldsymbol{f}}\|_{\mathcal{H}}$. Taking the dot product with $\boldsymbol{f}^* - \widehat{\boldsymbol{f}}$ and manipulating we obtain:

$$\mathcal{E}(\widehat{\boldsymbol{f}} - \boldsymbol{f}^*) = \frac{\sigma}{n}\sum_{i=1}^{n}w_i(\widehat{\boldsymbol{f}}(\boldsymbol{x}_i) - \boldsymbol{f}^*(\boldsymbol{x}_i)) + \delta\|\widehat{\boldsymbol{f}}\|_{\mathcal{H}}\frac{1}{n}\sum_{i}(\boldsymbol{f}^*(\boldsymbol{x}_i) - \widehat{\boldsymbol{f}}(\boldsymbol{x}_i))\widehat{z}_i$$

$$+ \delta\left(\frac{1}{n}\sum_{i}|y_i - \langle\widehat{\boldsymbol{f}}, \boldsymbol{\phi}(\boldsymbol{x}_i)\rangle| + \delta\|\widehat{\boldsymbol{f}}\|_{\mathcal{H}}\right)\langle\widehat{\boldsymbol{w}}, \boldsymbol{f}^* - \widehat{\boldsymbol{f}}\rangle.$$

Next, we bound each of the terms highlighted above one by one. First:

$$\langle\widehat{\boldsymbol{w}}, \boldsymbol{f}^* - \widehat{\boldsymbol{f}}\rangle \stackrel{(a)}{=} \langle\widehat{\boldsymbol{w}}, \boldsymbol{f}^*\rangle - \|\widehat{\boldsymbol{f}}\|_{\mathcal{H}} \stackrel{(b)}{\leq} \|\boldsymbol{f}^*\|_{\mathcal{H}} - \|\widehat{\boldsymbol{f}}\|_{\mathcal{H}}$$

Where (a) uses that $\widehat{\boldsymbol{w}}^{\top}\widehat{\boldsymbol{f}} = \|\widehat{\boldsymbol{f}}\|_{\mathcal{H}}$ which follows by the definition of subderivative. And (b) follows Cauchy–Schwarz inequality $\langle\widehat{\boldsymbol{w}}, \boldsymbol{f}^*\rangle \leq \|\widehat{\boldsymbol{w}}\|_{\mathcal{H}}\|\boldsymbol{f}^*\|_{\mathcal{H}} \leq \|\boldsymbol{f}^*\|_{\mathcal{H}}$. Moreover,

$$0 \leq \frac{1}{n}\sum_{i}|y_i - \langle\widehat{\boldsymbol{f}}, \boldsymbol{\phi}(\boldsymbol{x}_i)\rangle| \leq \frac{1}{n}\sum_{i}|\boldsymbol{f}^*(\boldsymbol{x}_i) - \widehat{\boldsymbol{f}}(\boldsymbol{x}_i)| + \frac{\sigma}{n}\sum_{i}|w_i| \leq \sqrt{\mathcal{E}(\widehat{\boldsymbol{f}} - \boldsymbol{f}^*)} + \sigma$$

Similarly, we have that:

$$\frac{1}{n}\sum_{i}(\boldsymbol{f}^*(\boldsymbol{x}_i) - \widehat{\boldsymbol{f}}(\boldsymbol{x}_i))\widehat{z}_i \leq \frac{1}{n}\sum_{i}|\boldsymbol{f}^*(\boldsymbol{x}_i) - \widehat{\boldsymbol{f}}(\boldsymbol{x}_i)| \leq \sqrt{\mathcal{E}(\widehat{\boldsymbol{f}} - \boldsymbol{f}^*)}$$

Now, we have two options on how to bound the first term in red:

**Bounds as a function of $\gamma$.** By the definition of $\gamma$ in the theorem assumption:

$$\frac{\sigma}{n}\sum_{i=1}^{n}w_i(\widehat{\boldsymbol{f}}(\boldsymbol{x}_i) - \boldsymbol{f}^*(\boldsymbol{x}_i)) \leq \sigma\gamma\|\boldsymbol{f}^* - \widehat{\boldsymbol{f}}\|_{\mathcal{H}}$$

Hence,

$$\mathcal{E}(\widehat{\boldsymbol{f}} - \boldsymbol{f}^*) \leq \sigma\gamma\|\boldsymbol{f}^* - \widehat{\boldsymbol{f}}\|_{\mathcal{H}} + \delta\|\widehat{\boldsymbol{f}}\|_{\mathcal{H}}\sqrt{\mathcal{E}(\widehat{\boldsymbol{f}} - \boldsymbol{f}^*)} + \delta\left(\sqrt{\mathcal{E}(\widehat{\boldsymbol{f}} - \boldsymbol{f}^*)} + \sigma + \delta\|\widehat{\boldsymbol{f}}\|_{\mathcal{H}}\right)(\|\boldsymbol{f}^*\|_{\mathcal{H}} - \|\widehat{\boldsymbol{f}}\|_{\mathcal{H}})$$

rearranging:

$$\mathcal{E}(\widehat{\boldsymbol{f}} - \boldsymbol{f}^*) \le \sigma(\gamma\|\boldsymbol{f}^* - \widehat{\boldsymbol{f}}\|_{\mathcal{H}} + \delta\|\boldsymbol{f}^*\|_{\mathcal{H}} - \delta\|\widehat{\boldsymbol{f}}\|_{\mathcal{H}}) + \delta\|\boldsymbol{f}^*\|_{\mathcal{H}}\sqrt{\mathcal{E}(\widehat{\boldsymbol{f}} - \boldsymbol{f}^*)} + \delta\|\widehat{\boldsymbol{f}}\|_{\mathcal{H}}(\delta\|\boldsymbol{f}^*\|_{\mathcal{H}} - \delta\|\widehat{\boldsymbol{f}}\|_{\mathcal{H}})$$

$$\le \sigma(\gamma\|\boldsymbol{f}^* - \widehat{\boldsymbol{f}}\|_{\mathcal{H}} + \delta\|\boldsymbol{f}^*\|_{\mathcal{H}} - \delta\|\widehat{\boldsymbol{f}}\|_{\mathcal{H}}) + \frac{\zeta}{2}\mathcal{E}(\widehat{\boldsymbol{f}} - \boldsymbol{f}^*) + \frac{1}{2\zeta}\delta^2\|\boldsymbol{f}^*\|_{\mathcal{H}}^2 + \delta^2\|\widehat{\boldsymbol{f}}\|_{\mathcal{H}}\|\boldsymbol{f}^*\|_{\mathcal{H}} - \delta^2\|\widehat{\boldsymbol{f}}\|_{\mathcal{H}}^2$$

$$\le \sigma(\gamma\|\boldsymbol{f}^* - \widehat{\boldsymbol{f}}\|_{\mathcal{H}} + \delta\|\boldsymbol{f}^*\|_{\mathcal{H}} - \delta\|\widehat{\boldsymbol{f}}\|_{\mathcal{H}}) + \frac{\zeta}{2}\mathcal{E}(\widehat{\boldsymbol{f}} - \boldsymbol{f}^*)$$
$$+ \delta^2\frac{1}{2\zeta}\left(\|\boldsymbol{f}^*\|_{\mathcal{H}} + \zeta(\kappa-1)\|\widehat{\boldsymbol{f}}\|_{\mathcal{H}}\right)\left(\|\boldsymbol{f}^*\|_{\mathcal{H}} - \zeta(\kappa+1)\|\widehat{\boldsymbol{f}}\|_{\mathcal{H}}\right)$$

where $\kappa^2 = 1 + 2/\zeta$. Now, let $\zeta = 2$ we have $\kappa = \sqrt{2}$ and

$$0 \le \sigma(\gamma\|\boldsymbol{f}^* - \widehat{\boldsymbol{f}}\|_{\mathcal{H}} + \delta\|\boldsymbol{f}^*\|_{\mathcal{H}} - \delta\|\widehat{\boldsymbol{f}}\|_{\mathcal{H}})$$
$$+ \frac{\delta^2}{4}\left(\|\boldsymbol{f}^*\|_{\mathcal{H}} + 2(\sqrt{2}-1)\|\widehat{\boldsymbol{f}}\|_{\mathcal{H}}\right)\left(\|\boldsymbol{f}^*\|_{\mathcal{H}} - 2(\sqrt{2}+1)\|\widehat{\boldsymbol{f}}\|_{\mathcal{H}}\right)$$

If $\|\widehat{\boldsymbol{f}}\|_{\mathcal{H}} \ge (1 + \frac{\gamma}{\delta})\|\boldsymbol{f}^*\|_{\mathcal{H}}$, the left hand side above is negative and we have a contradiction. Therefore, the result holds. $\qquad\square$

**Bounds as a function of $\beta$.**    Alternatively, by the definition of $\beta$:

$$\color{red}{\frac{\sigma}{n}\sum_{i=1}^n w_i(\widehat{\boldsymbol{f}}(\boldsymbol{x}_i) - \boldsymbol{f}^*(\boldsymbol{x}_i)) \le \sigma\beta\sqrt{\mathcal{E}(\widehat{\boldsymbol{f}} - \boldsymbol{f}^*)}.}$$

A completely equivalent derivation should yield that:

$$\mathcal{E}(\widehat{\boldsymbol{f}} - \boldsymbol{f}^*) \le \sigma\beta\sqrt{\mathcal{E}(\widehat{\boldsymbol{f}} - \boldsymbol{f}^*)} + \sigma(\delta\|\boldsymbol{f}^*\|_{\mathcal{H}} - \delta\|\widehat{\boldsymbol{f}}\|_{\mathcal{H}}) + \frac{\zeta}{2}\mathcal{E}(\widehat{\boldsymbol{f}} - \boldsymbol{f}^*)$$
$$+ \delta^2\frac{1}{2\zeta}\left(\|\boldsymbol{f}^*\|_{\mathcal{H}} + \zeta(\kappa-1)\|\widehat{\boldsymbol{f}}\|_{\mathcal{H}}\right)\left(\|\boldsymbol{f}^*\|_{\mathcal{H}} - \zeta(\kappa+1)\|\widehat{\boldsymbol{f}}\|_{\mathcal{H}}\right)$$

If we set $\zeta = 1$ we obtain $\kappa = \sqrt{3}$ and therefore:

$$\frac{1}{2}\mathcal{E}(\widehat{\boldsymbol{f}} - \boldsymbol{f}^*) \le \sigma\beta\sqrt{\mathcal{E}(\widehat{\boldsymbol{f}} - \boldsymbol{f}^*)} + \sigma(\delta\|\boldsymbol{f}^*\|_{\mathcal{H}} - \delta\|\widehat{\boldsymbol{f}}\|_{\mathcal{H}})$$
$$+ \delta^2\frac{1}{2}\left(\|\boldsymbol{f}^*\|_{\mathcal{H}} + (\sqrt{3}-1)\|\widehat{\boldsymbol{f}}\|_{\mathcal{H}}\right)\left(\|\boldsymbol{f}^*\|_{\mathcal{H}} - (\sqrt{3}+1)\|\widehat{\boldsymbol{f}}\|_{\mathcal{H}}\right)$$

Applying Proposition 5 to the first term :

$$0 \le \sigma^2\beta^2 + \sigma(\delta\|\boldsymbol{f}^*\|_{\mathcal{H}} - \delta\|\widehat{\boldsymbol{f}}\|_{\mathcal{H}}) + \delta^2\frac{1}{2}\left(\|\boldsymbol{f}^*\|_{\mathcal{H}} + (\sqrt{3}-1)\|\widehat{\boldsymbol{f}}\|_{\mathcal{H}}\right)\left(\|\boldsymbol{f}^*\|_{\mathcal{H}} - (\sqrt{3}+1)\|\widehat{\boldsymbol{f}}\|_{\mathcal{H}}\right)$$

If $\delta\|\widehat{\boldsymbol{f}}\|_{\mathcal{H}} \ge \delta\|\boldsymbol{f}^*\|_{\mathcal{H}} + \sigma\beta^2$ the left-hand side above is negative and we have a contradiction. Therefore,

$$\delta\|\widehat{\boldsymbol{f}}\|_{\mathcal{H}} \le \delta\|\boldsymbol{f}^*\|_{\mathcal{H}} + \sigma\beta^2 \overset{(a)}{\le} \delta\|\boldsymbol{f}^*\|_{\mathcal{H}} + \sigma\beta.$$

Where we use in (a) that $\beta \le 1$, since:

$$\frac{1}{n}\sum_{i=1}^n w_i\boldsymbol{g}(\boldsymbol{x}_i) \le \|\boldsymbol{w}\|\sqrt{\frac{1}{n}\sum_{i=1}^n \boldsymbol{g}(\boldsymbol{x}_i)^2} \le n\sqrt{\mathcal{E}(\boldsymbol{g})}.$$

### B.3    Proof of Proposition 3

The following theorem is useful for obtaining probabilistic bounds it is proved in (Wainwright, 2019, Theorem 2.26) .

**Theorem 3** (Lipschitz functions of Gaussian Variables (Wainwright, 2019)). Let $(X_1, \ldots, X_n)$ be a vector of i.i.d. standard Gaussian variables, and let $h : \mathbb{R}^n \to \mathbb{R}$ be $L$-Lipschitz with respect to the Euclidean norm. Then the variable $h(X) - \mathbb{E}[h(X)]$ is sub-Gaussian with parameter at most $L$, and hence

$$\mathbb{P}\left[|h(X) - \mathbb{E}[h(X)]| \geq t\right] \leq 2e^{-\frac{t^2}{2L^2}} \quad \text{for all } t \geq 0.$$

Now, lets start with the first statement. Lets write:

$$\gamma(\boldsymbol{w}) = \sup_{\|\boldsymbol{g}\|_{\mathcal{H}} \leq 1} \frac{1}{n} \sum_{i=1}^{n} w_i \boldsymbol{g}(\boldsymbol{x}_i) \tag{S.7}$$

notice that:

$$\frac{1}{n} \sum_{i=1}^{n} w_i(\boldsymbol{g}(\boldsymbol{x}_i)) = \frac{1}{n} \sum_{i=1}^{n} w_i \langle \boldsymbol{g}, \phi(\boldsymbol{x}_i) \rangle \leq \frac{1}{n} \|\boldsymbol{g}\|_{\mathcal{H}} \| \sum_i w_i \phi(\boldsymbol{x}_i) \|_{\mathcal{H}} \leq \frac{\sqrt{\lambda_{\max}}}{n} \|\boldsymbol{g}\|_{\mathcal{H}} \|\boldsymbol{w}\|_2$$

Where we used that:

$$\| \sum_i w_i \phi(\boldsymbol{x}_i) \|_{\mathcal{H}}^2 = \langle \sum_i w_i \phi(\boldsymbol{x}_i), \sum_j w_j \phi(\boldsymbol{x}_j) \rangle = \frac{1}{n} \sum_{i=1}^{n} \sum_j w_i w_j \langle \phi(\boldsymbol{x}_i), \phi(\boldsymbol{x}_i) \rangle = \boldsymbol{w}^\top K \boldsymbol{w} \leq \lambda_{\max} \|\boldsymbol{w}\|_2^2$$

hence $\gamma(\boldsymbol{w})$ is Lipschitz with constant $\frac{\sqrt{\lambda_{\max}}}{n} \leq \frac{\sqrt{\operatorname{tr}K}}{n} = \bar{\gamma}$. Then by Theorem 3:

$$P[\gamma > 2\bar{\gamma}] \leq 2\exp(-n^2 \bar{\gamma}/(2\lambda_{\max}))$$

The secons statement we follows from the follows directly from the following result

**Proposition 7.** If $\boldsymbol{w} \sim N(0, I_n)$ and $\epsilon > 0$. Then, for every $\boldsymbol{g} \in \mathcal{H}$ where $\sqrt{\mathcal{E}(\boldsymbol{g})} \geq \bar{\beta} + \epsilon$ we have:

$$\frac{1}{n} \sum_i w_i(\boldsymbol{g}(\boldsymbol{x}_i)) \leq (\bar{\beta} + \epsilon)\sqrt{\mathcal{E}(\boldsymbol{g})}$$

with probability $1 - \exp(-\frac{n(\bar{\beta}+\epsilon)}{2})$

The proof is given in (Wainwright, 2019, Lemma 13.12).

## C  Gaussian and local Gaussian complexities across different RKHS

### C.1  Computing the Gaussian complexity $\bar{\gamma}$.

Let

$$\bar{\gamma} = \mathbb{E}_{\boldsymbol{w} \sim \mathcal{N}(0, I_n)} \left[ \sup_{\|\boldsymbol{g}\|_{\mathcal{H}} \leq 1} \frac{1}{n} \sum_{i=1}^{n} w_i \, \boldsymbol{g}(\boldsymbol{x}_i) \right],$$

where $\mathcal{H}$ is a reproducing kernel Hilbert space (RKHS) with kernel $k(\cdot, \cdot)$ and corresponding Gram matrix $K \in \mathbb{R}^{n \times n}$ defined by $K_{ij} = k(\boldsymbol{x}_i, \boldsymbol{x}_j)$. By the reproducing property, $\boldsymbol{g}(\boldsymbol{x}_i) = \langle \boldsymbol{g}, k(\cdot, \boldsymbol{x}_i) \rangle_{\mathcal{H}}$, so that

$$\frac{1}{n} \sum_{i=1}^{n} w_i \, \boldsymbol{g}(\boldsymbol{x}_i) = \left\langle \boldsymbol{g}, \frac{1}{n} \sum_{i=1}^{n} w_i \, k(\cdot, \boldsymbol{x}_i) \right\rangle_{\mathcal{H}}.$$

Applying the Cauchy–Schwarz inequality, the supremum over $\|\boldsymbol{g}\|_{\mathcal{H}} \leq 1$ gives

$$\sup_{\|\boldsymbol{g}\|_{\mathcal{H}} \leq 1} \frac{1}{n} \sum_{i=1}^{n} w_i \, \boldsymbol{g}(\boldsymbol{x}_i) = \frac{1}{n} \left\| \sum_{i=1}^{n} w_i \, k(\cdot, \boldsymbol{x}_i) \right\|_{\mathcal{H}}.$$

Thus,

$$\bar{\gamma} = \frac{1}{n} \mathbb{E}_{\boldsymbol{w} \sim \mathcal{N}(0, I_n)} \left\| \sum_{i=1}^{n} w_i \, k(\cdot, \boldsymbol{x}_i) \right\|_{\mathcal{H}} = \frac{1}{n} \mathbb{E}_{\boldsymbol{w} \sim \mathcal{N}(0, I_n)} \sqrt{\boldsymbol{w}^\top K \, \boldsymbol{w}}.$$

Using Jensen's inequality and $\mathbb{E}[\boldsymbol{w}^\top K \boldsymbol{w}] = \mathrm{tr}(K)$, we obtain the bound

$$\bar{\gamma} \leq \frac{1}{n}\sqrt{\mathbb{E}[\boldsymbol{w}^\top K \boldsymbol{w}]} = \frac{\sqrt{\mathrm{tr}(K)}}{n}.$$

**Translational invariant kernel.** For translational invariant kernels $\sqrt{\mathrm{tr}(K)} = n$ and $\bar{\gamma} \leq 1/\sqrt{n}$.

**Linear kernel.** The linear kernel is defined here as:

$$k(\boldsymbol{x}, \boldsymbol{y}) = \boldsymbol{x}^\top \boldsymbol{y}$$

We have:

$$\bar{\gamma} = \frac{\sqrt{\mathrm{tr}\boldsymbol{X}\boldsymbol{X}^\top}}{n} = \frac{\sqrt{\sum_{i=1}^n \|\boldsymbol{x}_i\|_2^2}}{n} \leq \frac{1}{\sqrt{n}}(\max_i \|x_i\|_2) \tag{S.8}$$

## C.2 Computing the local Gaussian complexity $\bar{\beta}^2$.

**Linear kernels.** For linear kernels, we have $\bar{\beta} = O(\sqrt{\frac{p}{n}})$ the details are given in Example 13.8, Wainwright, 2019.

**Matérn kernels.** Here, we follow an argument similar to Wainwright, 2019, Example 13.20. We consider the following result, proved in Wainwright, 2019, Lemma 13.22, which provides a bound for a slightly different definition of $\bar{\beta}$. For simplicity, we adopt it informally—the main goal is to convey the dimensional dependence, and the necessary modifications to adapt it to our setting should follow naturally.

> **Proposition 8.** Consider an RKHS with kernel function $k$. For a given set of design points $\boldsymbol{x}_i$. And let $\mu_1 \geq \cdots \geq \mu_n \geq 0$ be he eigenvalues of $\frac{K}{n}$, where $K$ is the kernel matrix with entries $K_{ij} = k(\boldsymbol{x}_i, \boldsymbol{x}_j)$. Then for all $\bar{\beta}^2 > 0$ we have
>
> $$\mathbb{E}\left[\sup_{\|\boldsymbol{g}\|_{\mathcal{H}} \leq 1, \mathcal{E}(\boldsymbol{g}) \leq \bar{\beta}^2} |\frac{1}{n}\sum_{i=1}^n w_i \boldsymbol{g}(\boldsymbol{x}_i)|\right] \leq \sqrt{\frac{2}{n}}\sqrt{\sum_{i=1}^n \min(\bar{\beta}^2, \mu_i)}$$
>
> where $w_i \sim \mathcal{N}(0,1)$ are i.i.d Gaussian variables.

Let $k$ be the smallest positive integer such that for which

$$ck^{-2\nu/p} \leq \bar{\beta}^2 \tag{S.9}$$

we have that $\mu_i \leq ci^{-2\nu/p}$ for some constant $c$, see Bach, 2025.

$$\sqrt{\frac{2}{n}}\sqrt{\sum_{i=1}^n \min(\bar{\beta}^2, \mu_i)} \leq \sqrt{\frac{2}{n}}\sqrt{\sum_{i=1}^n \min(\bar{\beta}^2, ci^{-2\nu/p})} \leq \sqrt{\frac{2}{n}}\sqrt{\bar{\beta}^2 k + c\sum_{i=k+1}^n i^{-2\nu/p}}$$

We upper bound the sum by an integral,

$$c\sum_{i=k+1}^n i^{-2\nu/p} \leq c\int_{k+1}^\infty t^{-2\nu/p}dt \leq ck^{-2\nu/p+1} \leq \bar{\beta}^2 k$$

and hence:

$$\sqrt{\frac{2}{n}}\sqrt{\sum_{i=1}^n \min(\bar{\beta}^2, \mu_i)} \leq 2\sqrt{\frac{k}{n}}\bar{\beta}$$

therefore, the critical inequality is satisfied for $\bar{\beta} \geq 2\sqrt{\frac{k}{n}}$, now since $c'\bar{\beta}^{-p/2\nu} \leq \sqrt{k}$ by (S.9), the inequality is satisfied for any $\bar{\beta} \geq c''n^{-1/(2+p/\nu)}$. Therefore $\bar{\beta} = O(n^{-2/(2+p/\nu)})$.

# D  Adversarial multiple kernel learning

The next proposition is the equivalent proposition 2

> **Proposition 9.** Let $\boldsymbol{f} = \frac{1}{n} \sum_{i=1}^{n} \boldsymbol{f}_i$ where $\boldsymbol{f}_i \in \mathcal{H}_i$, then :
> $$\max_{\boldsymbol{d} \in (\cap_i \Omega_{\mathcal{H}_i})} (y - \langle \boldsymbol{f}, \boldsymbol{\phi}(\boldsymbol{x}) + \boldsymbol{d} \rangle)^2 = (|y - \sum_i \boldsymbol{f}_i(\boldsymbol{x})| + \delta \sum_j \|\boldsymbol{f}_j\|_{\mathcal{H}_j})^2.$$

A similar algorithm to that proposed in Algorithm 1 also apply for the case of adversarial multiple kernel learning. The algorithm also follows from the $\eta$-trick that yields the following variational reformulation of the loss:

$$(|y - \sum_j \boldsymbol{f}_j(\boldsymbol{x})| + \delta \sum_j \|\boldsymbol{f}_j\|_{\mathcal{H}_j})^2 = \min_{\eta^0, \eta^1} \frac{|y - \sum_j \boldsymbol{f}_j(\boldsymbol{x})|^2}{\eta^0} + \sum_j \frac{\delta \|\boldsymbol{f}_j\|_{\mathcal{H}_j}^2}{\eta^j}$$

$$\text{s.t. } \sum_j \eta^j = 1, \eta^j > 0.$$

We apply the above reformulation for each sample, obtaining and solve it as a block-wise coordinate problem, alternating between 1) solving over $\eta$ to compute the weights $w_i = 1/\eta_i^0$, $\lambda_i = \sum_i \frac{\delta}{\eta_i^1}$; and 2) solving the reweighted ridge regression problems that arises.

# E  Numerical experiments

## E.1  Datasets

We used 5 different data sets in our experiments.

- **Diabetes:** (Efron et al., 2004) The dataset has $p=10$ baseline variables (age, sex, body mass index, average blood pressure, and six blood serum measurements), which were obtained for $n=442$ diabetes patients. The model output is a quantitative measure of the disease progression.
- **Abalone:** (OpenML ID=30, UCI ID=1) Predicting the age of abalone from $p=8$ physical measurements. The age of abalone is determined by cutting the shell through the cone, staining it, and counting the number of rings through a microscope – a boring and time-consuming task. Other measurements, which are easier to obtain, are used to predict the age of the abalone. It considers $n=4417$ examples.
- **Wine quality:** (Cortez et al., 2009, UCI ID=186) A large dataset ($n=4898$) with white and red " vinho verde" samples (from Portugal) used to predict human wine taste preferences. It considers $p=11$ features that describe physicochemical properties of the wine.
- **Polution:** (McDonald & Schwing, 1973, OpenML ID=542) Estimates relating air pollution to mortality. It considers $p=15$ features including precipitation, temperature over the year, percentage of the population over 65, besides socio-economic variables, and concentrations of different compounds in the air. In total it considers It tries to predict age-adjusted mortality rate in $n=60$ different locations.
- **US crime:** (Redmond & Baveja, 2002, OpenML ID=42730, UCI ID=182) This dataset combines $p=127$ features that come from socio-economic data from the US Census, law enforcement data from the LEMAS survey, and crime data from the FBI for $n=1994$ comunities. The task is to predict violent crimes per capita in the US as target.

## E.2 Additional results

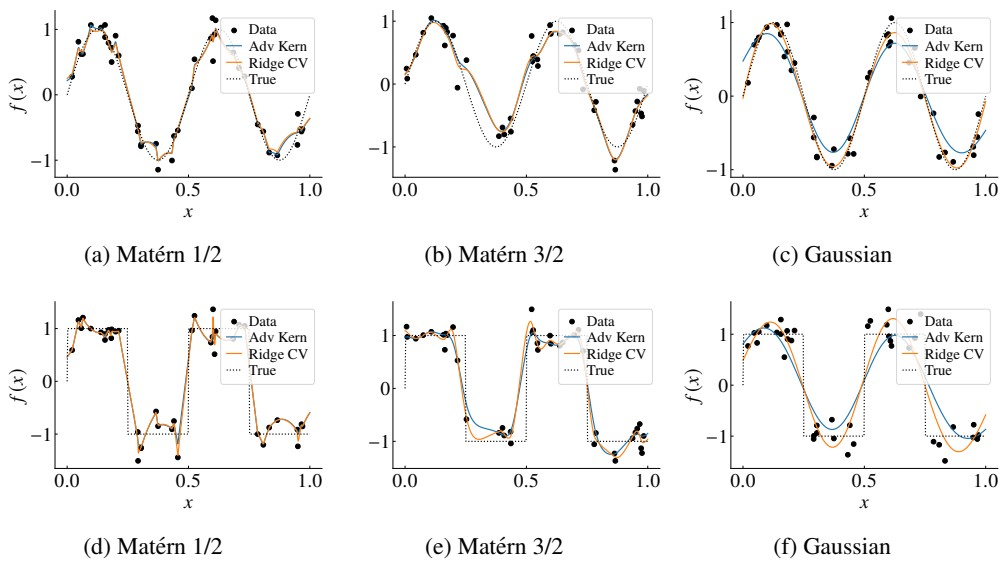

Figure S.1: We compare adversarial kernel training with cross-validated kernel ridge regression (Ridge CV). For (a)-(c), we show how it fits a smooth target function. For (d)-(f), we show how it fits a non-smooth target function.

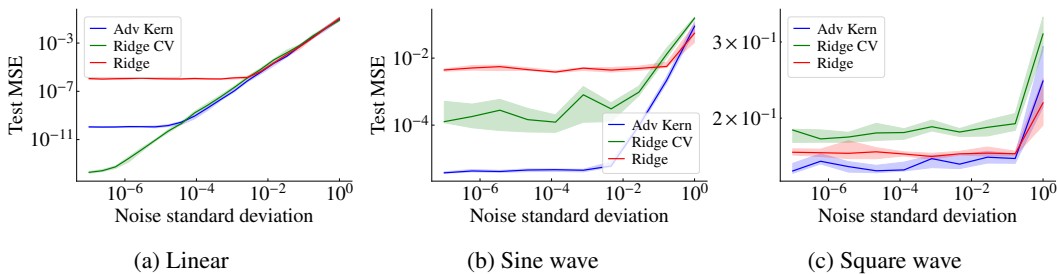

Figure S.2: We compare Adversarial Kernel Training (Adv Kern) against two baseline methods: Kernel Ridge Regression with cross-validation (Ridge CV) and Kernel Ridge Regression without cross-validation (Ridge). Our evaluation includes the sine and square wave examples shown in Figure S.1, using a Gaussian kernel, as well as a synthetic linear dataset using a linear kernel. To assess robustness to noise, we vary the standard deviation of Gaussian noise added to the data from $10^{-7}$ to $10^0$.

Table S.1: Empirically observed rate of convergence $r$, such that, Test MSE $\propto n^r$. Computed as the slope of the dashed line the linear approximation in Figure 1(right) and Figure 3.

|  | Non-smooth | | Smooth | |
|  | Adv Kern | Ridge CV | Adv Kern | Ridge CV |
|---|---|---|---|---|
| Matérn $1/2$ | -0.67 | -0.73 | -0.92 | -1.02 |
| Matérn $3/2$ | -0.45 | -0.73 | -1.06 | -1.07 |
| Matérn $5/2$ | -0.37 | -0.63 | -1.08 | -1.08 |
| Gaussian | -0.13 | -0.24 | -1.04 | -1.14 |

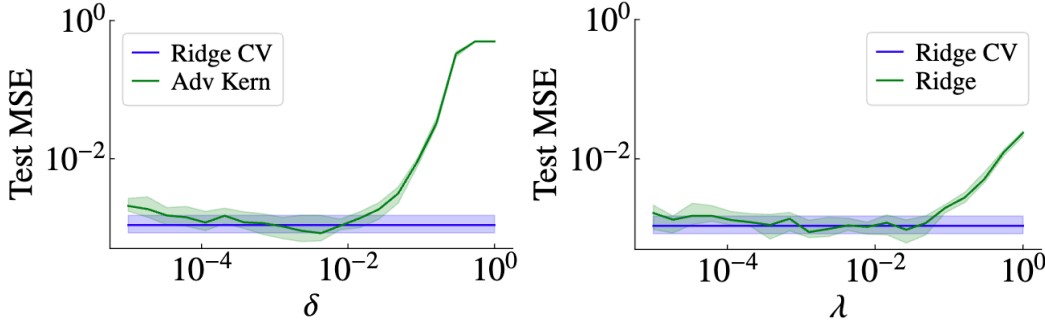

Figure S.3: We compare Adversarial Kernel Training (Adv Kern) and Kernel Ridge Regression without cross-validation (Ridge) against Kernel Ridge Regression with cross-validation (Ridge CV). We train and evaluate on the sine wave example shown in Figure S.1, using a Gaussian kernel, and plot the test MSE as a function of $\delta$ and $\lambda$ respectively.

| training method | no attack | test-time attack $\ell_2$ $\{\|\Delta\boldsymbol{x}\|_2 \leq 0.01\}$ | $\{\|\Delta\boldsymbol{x}\|_2 \leq 0.1\}$ | test-time attack $\ell_\infty$ $\{\|\Delta\boldsymbol{x}\|_\infty \leq 0.01\}$ | $\{\|\Delta\boldsymbol{x}\|_\infty \leq 0.1\}$ |
|---|---|---|---|---|---|
| **Adv Kern** ($\delta \propto n^{-1/2}$) | 0.70 (0.58–0.77) | 0.69 (0.59–0.76) | 0.66 (0.56–0.75) | 0.69 (0.57–0.76) | **0.58** (0.44–0.66) |
| Ridge Kernel ($\lambda$ cross-validated) | 0.68 (0.57–0.73) | 0.67 (0.56–0.73) | 0.62 (0.47–0.68) | 0.66 (0.54–0.71) | 0.49 (0.33–0.56) |
| **Adv Kern** $\{\|\boldsymbol{d}\|_{\mathcal{H}} \leq 0.01\}$ | **0.72** (0.59–0.84) | **0.72** (0.60–0.84) | **0.67** (0.51–0.78) | **0.71** (0.55–0.83) | 0.51 (0.29–0.63) |
| **Adv Kern** $\{\|\boldsymbol{d}\|_{\mathcal{H}} \leq 0.1\}$ | 0.67 (0.57–0.72) | 0.67 (0.58–0.72) | 0.65 (0.54–0.70) | 0.67 (0.57–0.72) | **0.58** (0.49–0.64) |
| Adv Input $\{\|\Delta\boldsymbol{x}\|_2 \leq 0.1\}$ | 0.67 (0.54–0.73) | 0.66 (0.55–0.72) | 0.61 (0.46–0.69) | 0.65 (0.51–0.72) | 0.47 (0.26–0.53) |
| Adv Input $\{\|\Delta\boldsymbol{x}\|_\infty \leq 0.1\}$ | 0.68 (0.57–0.75) | 0.67 (0.56–0.74) | 0.63 (0.51–0.69) | 0.66 (0.53–0.74) | 0.49 (0.30–0.56) |

(a) Pollution

| training method | no attack | test-time attack $\ell_2$ $\{\|\Delta\boldsymbol{x}\|_2 \leq 0.01\}$ | $\{\|\Delta\boldsymbol{x}\|_2 \leq 0.1\}$ | test-time attack $\ell_\infty$ $\{\|\Delta\boldsymbol{x}\|_\infty \leq 0.01\}$ | $\{\|\Delta\boldsymbol{x}\|_\infty \leq 0.1\}$ |
|---|---|---|---|---|---|
| **Adv Kern** ($\delta \propto n^{-1/2}$) | **0.38** (0.29–0.44) | 0.37 (0.26–0.43) | **0.31** (0.21–0.38) | 0.36 (0.27–0.43) | 0.20 (0.09–0.28) |
| Ridge Kernel ($\lambda$ cross-validated) | **0.38** (0.27–0.44) | 0.37 (0.26–0.43) | 0.29 (0.18–0.38) | 0.35 (0.25–0.43) | 0.15 (0.02–0.24) |
| **Adv Kern** $\{\|\boldsymbol{d}\|_{\mathcal{H}} \leq 0.01\}$ | **0.38** (0.27–0.44) | **0.38** (0.28–0.44) | **0.31** (0.19–0.37) | **0.37** (0.26–0.43) | **0.19** (0.09–0.27) |
| **Adv Kern** $\{\|\boldsymbol{d}\|_{\mathcal{H}} \leq 0.1\}$ | 0.32 (0.22–0.38) | 0.32 (0.23–0.38) | 0.27 (0.18–0.34) | 0.31 (0.22–0.37) | 0.17 (0.06–0.23) |
| Adv Input $\{\|\Delta\boldsymbol{x}\|_2 \leq 0.1\}$ | 0.37 (0.27–0.45) | 0.37 (0.25–0.43) | 0.29 (0.18–0.36) | 0.35 (0.24–0.43) | 0.15 (0.02–0.24) |
| Adv Input $\{\|\Delta\boldsymbol{x}\|_\infty \leq 0.1\}$ | 0.37 (0.26–0.45) | 0.37 (0.25–0.44) | 0.29 (0.19–0.38) | 0.35 (0.24–0.42) | 0.16 (0.04–0.22) |

(b) Diabetes

| training method | no attack | test-time attack $\ell_2$ $\{\|\Delta\boldsymbol{x}\|_2 \leq 0.01\}$ | $\{\|\Delta\boldsymbol{x}\|_2 \leq 0.1\}$ | test-time attack $\ell_\infty$ $\{\|\Delta\boldsymbol{x}\|_\infty \leq 0.01\}$ | $\{\|\Delta\boldsymbol{x}\|_\infty \leq 0.1\}$ |
|---|---|---|---|---|---|
| **Adv Kern** ($\delta \propto n^{-1/2}$) | **0.65** (0.63–0.67) | **0.65** (0.63–0.67) | **0.63** (0.61–0.65) | **0.63** (0.61–0.65) | **0.44** (0.41–0.47) |
| Ridge Kernel ($\lambda$ cross-validated) | **0.65** (0.63–0.67) | **0.65** (0.62–0.67) | 0.61 (0.59–0.63) | 0.62 (0.59–0.64) | 0.21 (0.16–0.24) |
| **Adv Kern** $\{\|\boldsymbol{d}\|_{\mathcal{H}} \leq 0.01\}$ | **0.65** (0.63–0.67) | **0.65** (0.63–0.67) | **0.63** (0.61–0.65) | **0.63** (0.61–0.66) | **0.43** (0.40–0.46) |
| **Adv Kern** $\{\|\boldsymbol{d}\|_{\mathcal{H}} \leq 0.1\}$ | 0.45 (0.44–0.46) | 0.45 (0.43–0.46) | 0.44 (0.42–0.45) | 0.44 (0.43–0.45) | 0.36 (0.35–0.37) |
| Adv Input $\{\|\Delta\boldsymbol{x}\|_2 \leq 0.1\}$ | **0.65** (0.63–0.67) | **0.65** (0.62–0.67) | 0.61 (0.59–0.63) | 0.62 (0.60–0.64) | 0.21 (0.17–0.24) |
| Adv Input $\{\|\Delta\boldsymbol{x}\|_\infty \leq 0.1\}$ | **0.65** (0.63–0.67) | **0.65** (0.63–0.67) | 0.61 (0.59–0.63) | 0.62 (0.59–0.64) | 0.21 (0.17–0.25) |

(c) US Crime

| training method | no attack | test-time attack $\ell_2$ $\{\|\Delta\boldsymbol{x}\|_2 \leq 0.01\}$ | $\{\|\Delta\boldsymbol{x}\|_2 \leq 0.1\}$ | test-time attack $\ell_\infty$ $\{\|\Delta\boldsymbol{x}\|_\infty \leq 0.01\}$ | $\{\|\Delta\boldsymbol{x}\|_\infty \leq 0.1\}$ |
|---|---|---|---|---|---|
| **Adv Kern** ($\delta \propto n^{-1/2}$) | 0.39 (0.37–0.40) | **0.38** (0.36–0.39) | **0.28** (0.26–0.29) | **0.36** (0.35–0.37) | 0.06 (0.04–0.07) |
| Ridge Kernel ($\lambda$ cross-validated) | 0.38 (0.36–0.39) | 0.36 (0.34–0.37) | 0.20 (0.18–0.21) | 0.33 (0.32–0.35) | -0.17 (-0.20—0.15) |
| **Adv Kern** $\{\|\boldsymbol{d}\|_{\mathcal{H}} \leq 0.01\}$ | **0.40** (0.39–0.41) | **0.38** (0.37–0.40) | 0.25 (0.23–0.26) | 0.36 (0.34–0.37) | -0.07 (-0.09—0.06) |
| **Adv Kern** $\{\|\boldsymbol{d}\|_{\mathcal{H}} \leq 0.1\}$ | 0.16 (0.16–0.17) | 0.16 (0.15–0.16) | 0.14 (0.14–0.15) | 0.16 (0.15–0.16) | **0.11** (0.10–0.11) |
| Adv Input $\{\|\Delta\boldsymbol{x}\|_2 \leq 0.1\}$ | 0.38 (0.36–0.39) | 0.36 (0.35–0.38) | 0.20 (0.19–0.22) | 0.33 (0.32–0.35) | -0.16 (-0.18—0.14) |
| Adv Input $\{\|\Delta\boldsymbol{x}\|_\infty \leq 0.1\}$ | 0.38 (0.36–0.39) | 0.36 (0.35–0.37) | 0.20 (0.19–0.21) | 0.33 (0.32–0.35) | -0.15 (-0.18—0.13) |

(d) Wine

| training method | no attack | test-time attack $\ell_2$ $\{\|\Delta\boldsymbol{x}\|_2 \leq 0.01\}$ | $\{\|\Delta\boldsymbol{x}\|_2 \leq 0.1\}$ | test-time attack $\ell_\infty$ $\{\|\Delta\boldsymbol{x}\|_\infty \leq 0.01\}$ | $\{\|\Delta\boldsymbol{x}\|_\infty \leq 0.1\}$ |
|---|---|---|---|---|---|
| **Adv Kern** ($\delta \propto n^{-1/2}$) | **0.57** (0.56–0.58) | **0.55** (0.54–0.57) | 0.39 (0.37–0.40) | **0.54** (0.52–0.55) | 0.13 (0.11–0.16) |
| Ridge Kernel ($\lambda$ cross-validated) | **0.57** (0.56–0.59) | **0.55** (0.53–0.56) | 0.26 (0.24–0.28) | 0.52 (0.51–0.54) | -0.16 (-0.20—0.13) |
| **Adv Kern** $\{\|\boldsymbol{d}\|_{\mathcal{H}} \leq 0.01\}$ | 0.56 (0.55–0.58) | **0.55** (0.53–0.56) | **0.40** (0.39–0.42) | 0.53 (0.52–0.55) | 0.18 (0.16–0.20) |
| **Adv Kern** $\{\|\boldsymbol{d}\|_{\mathcal{H}} \leq 0.1\}$ | 0.38 (0.37–0.39) | 0.38 (0.37–0.39) | 0.35 (0.34–0.36) | 0.37 (0.36–0.38) | **0.30** (0.29–0.31) |
| Adv Input $\{\|\Delta\boldsymbol{x}\|_2 \leq 0.1\}$ | **0.57** (0.56–0.59) | **0.55** (0.53–0.56) | 0.27 (0.25–0.29) | 0.52 (0.51–0.54) | -0.14 (-0.17—0.11) |
| Adv Input $\{\|\Delta\boldsymbol{x}\|_\infty \leq 0.1\}$ | **0.57** (0.55–0.58) | 0.54 (0.53–0.56) | 0.27 (0.25–0.29) | 0.52 (0.51–0.53) | -0.11 (-0.14—0.08) |

(e) Abalone

Table S.2: $R^2$ scores (higher is better) across datasets for varying adversarial training perturbation norms $p$ and adversarial radius $\delta$. We show the interquartile range obtained through bootstrap.

