# OpenReview forum: "Kernel Learning with Adversarial Features: Numerical Efficiency and Adaptive Regularization"
_NeurIPS.cc/2025/Conference — NeurIPS 2025 poster_

### Official Review · Reviewer_KwLt · 2025-06-14

**Clarity:** 2
**Significance:** 4
**Originality:** 4
**Rating:** 4
**Confidence:** 3

**Summary:**

This paper presents a new method for adversarial training in the regenerated kernel Hilbert space. By transferring perturbations from the input space to the feature space, convex optimization and adaptive regularization are achieved. Theoretical analysis and experiments verified the validity of the method.

**Questions:**

1.The comparison of related jobs (such as TRADES, etc.) is not comprehensive enough, and the Adversarial training methods in recent years have not been fully discussed in Adversarial training in deep learning, which weakening the positioning of the methods in the field of deep learning

2.In the introduction, a comparison of non-RKHS methods (such as adversarial training in deep learning) is added to highlight the theoretical uniqueness of the method proposed in this paper.

3. The inheritance and development relationship with the work of Ribeiro et al. (2023) can be clearly stated to avoid repetition.

4. If only PGD attacks are tested, why not verify the robustness against other attacks (such as FGSM, C&W)?

5. How sensitive δ and λ are to the results, and the hyperparameter selection criteria (such as the specific implementation effect of cross-validation) are not specified. The influence of the ε value in η-trick on the convergence rate and stability has not been fully discussed.

6. The alternating use of "adversarial perturbation" and "feature perturbation" can easily lead to misunderstandings. It is recommended to review the entire text and improve the manuscript.

7. “As shown in Figure 1...” However, the content of Figure 1 was not explicitly mentioned in the paper.

**Ethical Concerns:**

["NO or VERY MINOR ethics concerns only"]

**Final Justification:**

Thank you for the authors' efforts during the rebuttal period, my main concern are resolved. I will maintain my score.

**Limitations:**

The paper is valuable in terms of methodological innovation and theoretical analysis, but it does not fully discuss the technical limitations and neglects the analysis of social impact, which may reduce its credibility in practical applications.

**Quality:**

3

**Strengths And Weaknesses:**

The overall framework is theoretically innovative, but the experimental design and writing expression need to be further improved.

1.The upper bound relationship of the characteristic perturbation objective function with respect to the input perturbation target was derived, and the generalization error limit was derived.

2.Adversarial training migrates from the input space to the feature space, solving the computational bottleneck of the traditional min-max problem.

3.Extend the framework to multiple kernel learning, and the mathematical proof is rather rigorous.

---

> ### Author Rebuttal · Authors · 2025-07-29
>
> Thank you very much for your constructive feedback. We answer your Questions **(Q)** below and will update the paper accordingly.
>
> * **(Q1 & Q2) On related literature on adversarial training for deep learning.** Thank you for this important observation. We agree that our current discussion of adversarial training methods in deep learning could be strengthened. Our framework aligns with recent efforts that aim to make adversarial training more efficient through simpler, theoretically grounded regularization techniques. Indeed, recent works have sought to simplify adversarial training by relying on single-step perturbations \[2\] or latent-space perturbations \[3\], rather than more costly multi-step procedures. As you mention, methods such as TRADES \[1\] explore variants of robust optimization that trade off accuracy and robustness using tractable approximations. Overall, we believe there is also a tendency to try to use fast and simpler methods that align with our work. We will revise the introduction and related work section to include a more comprehensive comparison with adversarial training in the deep learning setting and highlight our contribution relative to these recent advances.
> * **(Q3)** **On more explicit mention of Ribeiro et al. (2023).** We agree and will make this connection more explicit to avoid unnecessary repetition, highlighting early on the links to Ribeiro et al. (2023) and how our work extends it to the nonparametric setting.
> * **(Q4) On the use of PGD.** Thank you for the comment. We will add a sentence to justify our choice. We use PGD because it is a strong multi-step adversary that generalizes many other methods. PGD was introduced by Madry et al. \[4\] as a unifying framework for first-order adversarial attacks (see Section 3, “Towards Universally Robust Networks”), offering a principled formulation for adversaries. Many popular attacks, such as FGSM and C\&W, can be interpreted as special cases of PGD with fewer steps or different constraints. This unified perspective facilitates a more general and consistent analysis, which may be obscured when relying on weaker, single-step methods.
> * **(Q5) On the influence of hyperparameters.** Thank you for the suggestion—this is indeed an important point. The performance variation with respect to $\\delta$ is highly relevant, and we will include additional analysis in the paper. The table below shows how the test MSE of Adversarial Kernel Regression changes as a function of $\\delta$. In this experiment, we used an RBF kernel and 100 training points sampled from a sine curve with additive Gaussian noise (standard deviation 0.1). For reference, standard kernel ridge regression with cross-validation achieves a test MSE of 0.001. We will include this and other sensitivity analyses in the revised version of the paper.
> | $\\delta$ | Test MSE |
> | :-----| ---- |
> | $1 \\times 10^{-5}$ | 0.002 |
> | $7 \\times 10^{-4}$ | 0.001 |
> | 0.026 | 0.002 |
> | 0.048 | 0.003 |
> | 0.086 | 0.009 |
> | 0.16 | 0.033 |
> | 0.30 | 0.329 |
> | 0.55 | 0.499 |
> | 1 | 0.499 |
>
>   About setting $\\epsilon$, we follow the recommendation of \[5\] and set $\\epsilon \= 1e-20$. Overall, our experience is that this is quite easy to set. Using a small value close to numerical precision yields good results, and the exact value is not important.
>
> * **(Q6)** **Using more consistent nomenclature.** Thank you for the suggestion. We agree and will revise the manuscript to use terminology more consistently, referring to our approach as "feature-space adversarial perturbation" throughout, to avoid confusion.
> * **(Q7) Mention all the figures in the main text.**  We agree. In the revised version, we will explicitly describe Figure 1 in the paragraph following it. So that the main text contains a description of each panel and the main takeaway that our method adapts to the function’s smoothness without requiring explicit tuning.
> * **(Limitations)** Thank you for the helpful and constructive feedback. We will incorporate the reviewer’s suggestions by clearly stating the main technical limitations in the revised version. While the paper is primarily theoretical, we agree it is important to reflect on societal impact and will add a brief discussion—particularly noting that safer, adversarially trained models often come with increased computational cost, which we hope can be mitigated through improved formulations such as the one we propose.
>
> **References:**
>
> * \[1\] "Theoretically Principled Trade-off between Robustness and Accuracy" H. Zhang, Y. Yu, J. Jiao, E. Xing, L. El Ghaoui, and M. Jordan. *ICML*, 2019
> * \[2\] "Fast is Better than Free: Revisiting Adversarial Training" E. Wong, L. Rice, and J. Z. Kolter. *ICLR*, 2020\.
> *  \[3\] "Reliably Fast Adversarial Training via Latent Adversarial Perturbation" G. Y. Park and S. W. Lee. *ICCV,* 2021\.
> *  \[4\] "Towards Deep Learning Models Resistant to Adversarial Attacks" A. Madry, A. Makelov, L. Schmidt, D. Tsipras, and A. Vladu. ICLR, 2018\.
> *  \[5\] "Efficient Optimization Algorithms for Linear Adversarial Training" A. H. Ribeiro, T. B. Schön, D. Zachariah, and F. Bach. *AISTATS*, 2025\.

---

### Official Review · Reviewer_PoeL · 2025-06-21

**Clarity:** 3
**Significance:** 2
**Originality:** 3
**Rating:** 4
**Confidence:** 3

**Summary:**

This paper looks at adversarial optimization in the context of adversarial robustness. Instead of framing the problem as a min-max optimization with respect to perturbation in input space, they consider perturbations in feature space, which allow to translate the min-max problem in a convex optimization algorithm with convergence guarantees. The authors later provide theoretical generalization bounds on this method, and provide numerical evidence that this method has advantages with respect to kernel ridge regression and standard robust training.

**Questions:**

Line 73-74: the Authors claim they consider square loss for simplicity and they point out that some developements can be applied more generally. Can they elaborate on this? Do not find an argument about this statement in the paper.

Line 155: if the Authors refer to the fact that in ridge regression the optimal regularization $\lambda$ depends on the noise, maybe adding one line of explanation, reference can improve the readability.

**Ethical Concerns:**

["NO or VERY MINOR ethics concerns only"]

**Final Justification:**

The final justification is in my discussion comment for AC and Reviewers.

**Limitations:**

It seems to me this paper does not have any theoretical insight for the high dimensional. If this is true, it is fair to discuss this.

The paper does not carry a clear connection between the meaning of $\delta$ in input space and on a generic feature space. In words, a fixed value of $\delta$ for a specific feature map, would provide robustness guarantees as if the optimization was like the one in Eq. (1) for which radius of $\Omega_X$?

**Quality:**

3

**Strengths And Weaknesses:**

**Strenghts**

The paper discusses what i think to be a rather original approach to robust optimization. Furthermore, the paper is well written and presented with rigor.

---

**Weaknesses**

The Authors minimally elaborate on what is the exact set $\Omega_{X}$ for different feature maps. They provide examples in Table 1, but I would in general stress that this connection is important to track the relevance of the the value $\delta$ to express robustness (which generally considers perturbations in input space, not feature space)

To my understanding, these results are meaningful in the small dimensional setting, i.e. the scalings are for sufficiently large number of samples $n$, and the number of input dimensions $p$ is much smaller (see for example Table 2). This should also be a necessary assumption to have a direct connection between the fixed and random design (see line 153), as overfitting here becomes negligible. Can the authors remark on this point? In case I am correct, given the high popularity of high dimensional learning in the recent literature I would strongly suggest to remark this limitation in the revision.

At line 178 I found myself confused: the Authors mention "optimal choices for $\delta$". To my understanding, $\delta$ is a parameter that guarantees more robustness as it grows larger, and performances are in general monotonically decreasing as $\delta$ increases (tradeoff between robustness and generalization). The narrative here seems to have a different angle, as $\delta$ is framed more as a hyper-parameter that enables regularization. Can the authors explain this difference in perspective? This is maybe my main concern regarding the significance of this work. This is also not clear with respect to the results shown in Table 3, where $\delta$ is directly associated with the resistance to some attack...

It is not totally clear to me how to parse the numerical experiments in the last section: in general, I would compare the test error of two algorithms for a fixed level of safety (which I identify as the resistance to adversarial perturbations). Here, a fixed value of $\delta$ gives different robustness guarantees depending on the specific kernel (as shown for few examples in Table 1). It is possible that the proposed method has better generalization simply because it is guaranteeing "less robustness" for input perturbations (I am referring to Figure 3 in this case). Considering my previous comment, it is possible I did not fully understood what is the role of $\delta$ the Authors want to stress here...

---

> ### Author Rebuttal · Authors · 2025-07-29
>
> Thank you very much for your comments—they were extremely helpful. You raised several important points that we aim to clarify in the revised version of the paper. Below, we respond to your specific questions, as well as the limitations (L) and weaknesses (W) you identified. For reference, we use labels such as L1 for the first point raised in the first paragraph of “Limitations,” W2 for the second point raised in the “Weaknesses” section, and so on.
>
> * **(L1/W2)   Importance of our analysis in the high-dimensional regime.** We want to highlight that the results from Section 5 of our paper are relevant in the high-dimensional regime. Particularly, they have strong parallels to that of Chapter 13 in the book “High-Dimensional Statistics,” \[3\] where similar bounds are used to analyse Kernel Ridge Regression in the high-dimensional regime.
>   * **About Table 2\.** We included the bounded $p$ assumption for the Gaussian kernel in Table 2 mainly for clarity, as it allowed us to provide a simple closed-form expression for $\\beta$. However, we agree this could be confusing, since all other results hold for arbitrary $p$. We will revise the text to make this distinction clearer and avoid potential misunderstanding.
>   * **Two different regimes.** We obtain two different regimes in Theorem 1\. One regime depends on $\\gamma$ and is independent of the input dimension. This bound remains relevant even in very high-dimensional settings. The second regime depends on $\\beta$, and for the linear and Matérn kernels, the dependency on the input dimension $p$ is made explicit in the expressions for $\\beta$, allowing direct analysis in high-dimensional scenarios. This bound can sometimes outperform the dimension-free one when $p$ is small, but it is typically looser in the high-dimensional regime.
>   * **Analysing overfitting.** Our analysis also sheds light on the bias–variance trade-off. The upper bounds we derive reveal an explicit dependence on the noise level $\\sigma$ and admit a natural interpretation: terms involving $R$ capture the bias component, while terms involving $\\sigma$ reflect variance. For example, when $\\delta \= 0$, the term $B^{\\text{adv}}$ recovers the classical bound for linear regression, $\\sigma^2 d / n$, illustrating how overfitting arises when the number of parameters is too large. In contrast, the bound $A^{\\text{adv}}$ remains independent of the number of parameters, emphasizing the regularizing effect of adversarial training.
> * **(L2/W1) Highlighting the exact meaning of set $\\Omega\_X$.**  Thank you a lot for the question about  $\\Omega\_X$. We will try to make things more explicit in the main text, but we highlight that Table 1 gives one to one equivalences between radius on the feature space  $\\delta\_{features}$ and the input space $\\delta\_{inp}$.  Under these choices, our method gives robustness equivalences that also hold in the input space. For instance for $\\delta\_{inp} \\Leftrightarrow  \\delta\_{features}$ for linear kernels;  $\\delta\_{inp} \\Leftrightarrow C\\delta\_{features}^{1/d}$ for polynomial and so on.  We will better highlight this connection, bringing some insight into the main text. We will also make clear the limitations of such analysis, particularly for the kernels where the $\\delta\_{inp}$  in the input space increases very slowly with  $\\delta\_{features}$ in the feature space.
> * **(W3)  Robustness vs regularization.** Thank you for raising this important point. We agree that the dual role of adversarial training is central to the paper, and we appreciate the opportunity to clarify this. In our work, we emphasize two distinct perspectives on adversarial training:
>    1. **Robustness perspective:** Adversarial training is used to defend against perturbations in the input space—larger values of $\\delta$ offer stronger protection but may degrade generalization.
>    2. **Regularization perspective:** Following Ribeiro et al.  \[4\], we view adversarial training as a form of regularization, where $\\delta$ acts as a tuning parameter akin to the regularization strength in ridge regression. A key benefit is that it can be chosen independently of the noise level, e.g., set as $\\delta \\propto 1/\\sqrt{n}$.  We will revise the text to clearly distinguish these two perspectives early in the paper and avoid confusion.
> * **(W3) Optimal choice of $\\delta$.** Indeed, there is a conflicting goal here on how to set $\\delta$ and a tradeoff between robustness and generalization. For robustness, $\\delta$ should remain fixed to guarantee resistance to a specific threat model. In contrast, from the generalization/regularization viewpoint, we allow $\\delta$ to decrease with sample size, enabling the estimator to approach the true function—much like a shrinking regularization path. When we refer to the "optimal choice" of $\\delta$, we are referring specifically to the second perspective (generalization). We will make this more explicit in the revision.
> * **(W4) Experiments with $\\delta$ constant.**  We will include experiments with a fixed $\\delta$. Below, we present a representative example using the *pollution* dataset. Robustness is evaluated against input-space attacks with $p=\\infty$ and $\\delta\_{\\text{inp}} \= 0.1$—a higher value than in the main tables, chosen to emphasize the effects.  Notably, our method outperforms direct input-space adversarial training under the same threat model ($p=\\infty$, $\\delta\_{\\text{inp}} \= 0.1$), especially in terms of robustness. Moreover, these results illustrate the trade-off between clean and adversarial accuracy: as the feature-space adversarial radius $\\delta$ increases, the model becomes more robust, albeit with a slight drop in performance on clean data. *We show the $R^2$ score (higher is better).*
> | $R^2$ score  | Clean data  |  Adversarial attack  |
> | :---- | ----: | ----: |
> | **Adv Kern** (ours, $\\delta\=0.01$) | **0.72** | 0.51 |
> | **Adv Kern** (ours, $\\delta\=0.1$) | 0.67 | **0.58** |
> | **Adv Inp** (same threat model as evaluation)  |  0.68 | 0.50 |
>
>
> * **(W4) Interpreting Figure 3\.** We agree that robustness to input perturbations depends on the kernel, since the same value of $\\delta$ in feature space implies different levels of robustness in input space, as shown in Table 1\. However, we stress that our primary goal in Figure 3 is to evaluate generalization performance, not robustness. To avoid confusion, we will revise the text to clarify the intended interpretation of each experiment and the role $\\delta$ plays in each setting.
> * **(Question about Line 73-74)** Thank you for the comment. We agree that this point deserves a clearer explanation in the text. Several of our developments can indeed be extended to classification settings. In particular, the core idea of adversarial training in feature space applies broadly, whether the task is regression or classification.
>  Proposition 1, which provides bounds on the input-feature perturbation relation, holds for classification as well. Proposition 2 also admits a natural analogue in the classification setting, under similar regularity assumptions. While the specific optimization algorithm we propose relies on the structure of the square loss, related approaches for classification have been developed—e.g., Ribeiro et al. \[1\] propose an algorithm in the linear setting that could be extended to the nonparametric case considered here.
>   Finally, Sections 5 and 6, which study generalization and empirical performance, could in principle be adapted to classification tasks. However, doing so would require further theoretical and experimental work, and we believe it is best left for future work. We will revise the text to clarify these points.
> * **(Question about Line 155\)** Thank you for the comment. You are absolutely right. We will add comments and a reference to improve readability.  More specifically, one place where this is shown is in Proposition 3.8 of  \[2\]
>
> **References:**
>
> * \[1\] "Efficient Optimization Algorithms for Linear Adversarial Training" A. H. Ribeiro, T. B. Schön, D. Zachariah, and F. Bach. *AISTATS*, 2025\.
> * \[2\] "Learning Theory from First Principles" F. Bach. *The MIT Press*, 2024\.
> * \[3\] "High-Dimensional Statistics: A Non-Asymptotic Viewpoint" M. J. Wainwright. *Cambridge University Press*, 2019\.
> * \[4\] “Regularization properties of adversarially trained linear regression,” A. H. Ribeiro,, D. Zachariah,  F. Bach, and T. B. Schön. *NeurIPS*, 2023

---

> > ### Comment · Reviewer_PoeL · 2025-08-03
> >
> > I thank the Authors for their careful rebuttal. I am still confused about few points, will go through them below.
> >
> > **L1/W2**: Maybe I am missing something. Who is $d$? As far as I understood the notation, the dimensionality of the data here is $p$, and there is no notation for the dimensionality of the feature space. Is this correct? Also, what do you mean by "_all other results hold for arbitrary $p$_"? I presume that in some regime (like $p = \omega(n)$ all the generalization bounds become vacuous (this is what I expect, for example, in linear regression). I am missing this interplay here: in which regimes of $p$ are your results meaningful ($\mathcal E < R^2$)? I expect the answer to this question to be positive only when $p = o(n)$. Is that correct?
> >
> > **L2/W1**: I agree that your Table 1 makes this connection, but in table 3 you compare a fixed value of $\delta$ for perturbation in input space and perturbations in feature space. How is this comparison fair? Why should I believe that your method offers better robust generalization with respect to training methods based on input perturbations? What am I missing here?
> >
> > **W3**: I believe these perspectives to be fundamentally different. In the robustness perspective, $\delta$ is not really an hyper-parameter to optimize, but rather a budget to stick to. In this setting, I believe the proposed method should be compared with other robust optimization methods on robust generalization tasks with a fixed (input space) value of $\delta$. Is there any experiment in the paper on this (it seems to me this is not done in the newly provided table in the rebuttal regarding **W4**)? Regarding the regularization perspective, I presume a fair comparison is present in Figure 2. Can the Authors confirm on this?
> >
> > **W4**: Finally, I am a bit confused by how the authors can set different values of $p$ In the experiment shown in Table 3. I apologize for not having asked this before. I realized that I am not fully grasping this setting when the Authors mentioned $p=\infty$ in their rebuttal: how can the input dimensionality of the pollution dataset be $\infty$? Maybe I am missing the notation here (see also my first point).

---

> ### Author Response · Authors · 2025-08-03
>
> Thank you for the follow-up. We clarify it below
>
> - **(L2/W1) Table 3 compares all methods under the same exact conditions.** While the training procedures differ for different rows— input-space adversarial training *(AdvInput) vs.* our feature-space adversarial training (*AdvKern, ours)* *vs.* kernel ridge regression *(RidgeKernel)*— the evaluation itself is uniform. In last four columns, ***we report the attacks in the input space for all the methods***. If anything, this could favor input-space methods *AdvInput* slightly, since they are trained under the same conditions used for testing. Our method still outperforms under this shared evaluation (see Table 3), which suggests improved robust generalization despite the potentially mismatched training perturbation. We believe most of it is due to having a convex optimization that can potentially avoid local minima.
> - **(W3) Table 3 evaluates robust generalization**. Table 3 evaluates all models under the ***same*** adversarial threat model — namely, what is being reported in the last four columns is the mean error under ***input-space attack in the test set*** (Hence it is evaluating robust generalization in the traditional sense). The same holds for the newly provided experiment, the difference between the Table 3 and the new experiment on how the training is done. In Table 3 $\\delta \\propto 1/\\sqrt{n}$ and in the newly provided  $\\delta$ is fixed. ***We also confirm that*** in the  Figure 2 indeed adopts a ***regularization perspective***, comparing standard generalization performance across different adversarial radii during training (same for the no-attack column of Table 3).
> - **(W4) On the definition of $p$.**  We believe the confusion arose due to our overloaded use of notation here. In Table 3,***the symbol \\( p \\) refers to the norm used in the adversarial attack***—specifically, the attack is defined as    $$ \\|\\Delta x\\|\_p \\leq \\delta. $$  For example, when $ p \= \\infty$ and $  \\delta \= 0.1$, this corresponds to  $$ \\|\\Delta x\\|\_\\infty \\leq 0.1. $$   We acknowledge that elsewhere in the paper, $p$ is used to denote the problem dimension, which understandably led to confusion. We apologize for this and will revise the notation in the final version to avoid ambiguity.
> - **(L1/W2) Dimension-free bounds.**  As mentioned, the bounds involving $\\gamma$ are dimension-free and remain non-vacuous even when $ p \= \\omega(n)$, provided $\\delta  \propto\\gamma $. These rates do not depend on the input dimension \\( p \\), which makes them particularly relevant in high-dimensional settings.
>   - ***This property is not unique to our method***—it also holds for (kernel) ridge regression. For example, see Section 3.6.1 of *"Learning Theory from First Principles"* by F. Bach, MIT Press, 2024\. In ridge regression, one can derive dimension-free generalization bounds that achieve the *slow rate* $\\mathcal{O}(n^{-1/2})$, as opposed to the *fast rate* $\\mathcal{O}(n^{-1})$ that may be attainable when $p \= o(n) $ in linear models.
>   - Our method enjoys similar regularization behavior, offering the same generalization guarantees as kernel ridge regression, but with less need for tuning. We also emphasize that these bounds assume the ***target function lies in the RKHS associated with the chosen kernel***. For example, when using the Matérn kernel, we implicitly assume the function has sufficient smoothness to belong to this corresponding RKHS. These comments help contextualize the strengths and limitations of the bounds.
>
> Again, we truly appreciate the feedback. We will make sure all these points are clarified in the text

---

> > ### Comment · Reviewer_PoeL · 2025-08-07
> >
> > Thank you for your reply!
> >
> > - **(L2/W1) / (W3)** Aha! Got it, thanks! Maybe remark this in the revision, if you think it could help readability.
> >
> > - **(W4)** Ok I understand now. Yes please fix this mismatch in the revision.
> >
> > - **(L1/W2)** Ok I am still a bit confused here. This $p$ is the number of input dimensions $d$ right? I will assume so. I have question that maybe would help me to understand this better: in high dimensional linear regression, if the labels follow the rule $y = x^\top \theta^* + z$, where $z$ is some label noise, $\theta^*$ is the ground-truth, and $x$ assume is isotropic Gaussian, than the ridge estimator $\hat \theta$ (for any regularizer $\lambda$) will have a test risk not lower than $Var(y)$ if $d = \omega(n)$. In words, in linear regression one needs at least as many samples as effective dimensions $d$ to be able to learn something (see perhaps [1]). Given this example, I am looking for a sanity check to see in which high dimensional regime your bounds break. I expect this to be for a scaling of $d$ at most as the one of $n$.
> >
> > In case you still believe your bounds work for $d = \omega(n)$, can you provide some intuition why here the story is different with respect to linear regression? Is this maybe related to the last point you made on the target function lying in the RKHS?
> >
> > [1] Qiyang Han & Xiaocong Xu, _The distribution of Ridgeless least squares interpolators_, 2023

---

> ### Author Response · Authors · 2025-08-07
>
> Thank you for letting us clarify. Let us consider the standard linear model: $y_i = x_i^\top \theta_* + \varepsilon_i, \text{with } \dim(x_i) = p,$ and $ \varepsilon_i \sim \mathcal{N}(0, \sigma^2) $.
> Let
> $$
> \widehat{\Sigma} = \frac{1}{n} \sum_{i=1}^n x_i x_i^\top
> $$
> denote the empirical covariance matrix of the inputs. To build intuition, let  illustrate with ridge regression, where the estimator is defined as:
> $$
> \widehat{\theta} = \arg\min_{\theta} \frac{1}{n} \sum_{i=1}^n (y_i - x_i^\top \theta)^2 + \lambda ||\theta||^2.
> $$
> And the corresponding **excess risk** is given by: (see [2], proposition 3.7)
> $$
> \text{Excess Risk} =
> \lambda^2 \theta_*^\top \widehat{\Sigma} (\widehat{\Sigma} + \lambda I)^{-2} \widehat{\Sigma} \theta_*+
> \frac{\sigma^2}{n} \operatorname{tr} \left[ \widehat{\Sigma}^2 (\widehat{\Sigma} + \lambda I)^{-2} \right].
> $$
>
> The first term can seen as the Bias ($B$) and the second term as the Variance ($V$). Let us now analyse case by case.
>
> ---
> - **Case 1: No regularization ($\lambda = 0$) and $p<n$**
>
> In this case, the bias term vanishes: $B = 0.$ And the variance term simplifies to:
> $$
> V = \frac{\sigma^2}{n} \operatorname{tr}\left[ \widehat{\Sigma}^2 \widehat{\Sigma}^{-2} \right] = \frac{\sigma^2}{n} \operatorname{tr}(I) = \frac{\sigma^2 p}{n},
> $$
> which matches the bound you mentioned. This expression holds as long as $ \widehat{\Sigma} $ is full-rank. And here $p<n$ is a necessary condition for it to hold.
>
> ---
>
> - **Case 2: Ridgeless overparametrized ($p>n$ and $\lambda \to 0^+ $)**
>
> This regime has received considerable attention recently due to its connection with double descent and benign overfitting. Han & Xu [1] study this setting and highlight some of the limitations of the ridgeless interpolator, including the failure of its implicit regularization in certain cases.
>
> We emphasize that ***this is not the regimen we are discussing***. These regimes usually benefit from a random design analysis (in contrast to the fixed design analysis we present here).  ***In our case, we have a regularization parameter and we adapt it accordingly to the problem.***
>
> ---
> - **Case 3: Positive regularization ($\lambda > 0$) adapting to the problem signal-to-noise ratio** *(here the analysis is the same for both $p\le n$ and for $p >n$)*
>
> When $ \lambda > 0 $, a standard arguments shows that:  (see [2], proposition 3.8)
>
> $$
> B \leq \frac{\lambda}{2} ||\theta_*||_2^2, \quad
> V \leq \frac{\sigma^2}{2\lambda n} \operatorname{tr}(\widehat{\Sigma}),
> $$
>
> Importantly, ***in Cases 1 and 2 above***, the variance term involves the quantity
> $
> \operatorname{tr}\left[ \widehat{\Sigma}^2 \widehat{\Sigma}^{-2} \right] = \operatorname{tr}[I] = p,
> $
> which introduces an **explicit dependence on the dimension**. In contrast, when $\lambda > 0 $, the variance depends only on $ \operatorname{tr}(\widehat{\Sigma}), $ which remains bounded, for instance, if the inputs are bounded.
>
>
> Moreover, if one can choose the regularization parameter adaptively — for instance,
> $$\lambda_* = \frac{\sigma  \operatorname{tr}(\widehat{\Sigma})^{1/2}       }{||\theta_* ||_2 \sqrt{n}},$$
> yields the bound
> $$\text{excess risk} \leq \frac{\sigma  ||\theta||_2 \operatorname{tr}(\widehat{\Sigma})^{1/2} }{\sqrt{n}}$$
> This further illustrates that regularization allows for dimension-independent learning guarantees in high-dimensional settings, as long as the trace of the covariance remains controlled. This yields the $O(n^{-1/2})$ rate that does not depend on the dimension that was mentioned in the last response.
>
>
>
> ---
> - **How the dependcy on the dimension can still appear**
>
> The bounds that we were referring to as ”dimension-free” depend on $||\theta||_2$.  For some data generation processes, **we could have**  $||\theta||_2$ **potentially growing with the number of parameters**. In the case of the non-parametric regression, that is what we referred to as the assumption that the function that generates the data belongs to the RKHS, otherwise, we would have to consider larger and larger norms to get the desired result.
>
> -------
> - **What is the difference between (kernel) ridge regression and  (kernel) adversarial learning?**
>
> We show that similar bounds hold for  (kernel) adversarial regression. But with the advantage that we can get similar rates without a regularization parameter proportional to   $ \sigma/ ||\theta_*||_2 $. This is our Theorem 1 in the paper. There, we give a similar dimension-free bound dependent on the Gaussian complexity $\gamma$, which we call it $A^{\text{adv}}$. We also give a bound $B^{\text{adv}}$ that should recover a similar $\sigma^2 p / n$, when the adversarial radius is too close to zero.
>
> ---
>  ### References:
> - [1] "The distribution of Ridgeless least squares interpolators", Qiyang Han & Xiaocong Xu, 2023
> - [2]  "Learning Theory from First Principles" F. Bach. The MIT Press, 2024.

---

> > ### Comment · Reviewer_PoeL · 2025-08-08
> >
> > Thank you for your answer. I see what you mean. I would like to remark that if the trace of the sample covariance is of constant order, and $|| \theta^* ||_2$ is also of constant order, than either the labels are vanishing in absolute value or the problem is "effectively" low dimensional (think for example at Gaussian distribution with covariance $\Sigma$ in the case of linear regression, and where with "effectively" low dimensional I mean that the spectral norm of $\Sigma$ becomes comparable with its trace).
> >
> > In other words, from my (maybe very personal) perspective, the trace of the covariance is something that might be expected to scale with the number of input dimensions in a "natural" setting. I am not going against your point, but I would suggest to eventually remark this in the revision, whenever you have rates that are strictly speaking dimension independent, but that could depend on something like these high dimensional objects.
> >
> > Nevertheless, I confirm my acceptance recommendation.

---

> > > ### Author Response · Authors · 2025-08-08
> > >
> > > We agree with your comments regarding the scaling and we will make sure to add a remark about it.
> > >
> > > Thank you for the interesting discussion and valuable feedback. We greatly appreciate the acceptance recommendation.

---

### Official Review · Reviewer_vuz8 · 2025-06-28

**Clarity:** 3
**Significance:** 3
**Originality:** 3
**Rating:** 5
**Confidence:** 4

**Summary:**

This paper studies adversarial kernel regression, where we aim at choosing the best function in an RKHS that is robust to worst-case perturbations in the input. The authors provide a relaxation of this problem by considering worst-case perturbations over the feature space, which allows the inner maximization to be solved in closed-form and leading to a convex minimization problem that can be solved with block-coordinate descent type algorithms. The authors also prove generalization bounds and demonstrate that unlike ridge regression, the optimal value of perturbation strength does not depend on noise level.

**Questions:**

1. As the original adversarial training loss (with perturbations in the input space) is a maximization over convex functions, it is itself a convex minimization problem. How would solving the original problem compare to solving the relaxed version in the feature space?

2. A relevant reference may be [1], where the authors study how to learn multi-index models in an adversarially robust manner with two-layer neural nets. The authors show that the first layer can be trained in a standard way, which leads to a data-adapted RKHS. In the second stage, they need to minimize the adversarial loss over this RKHS. The approach used in this paper can be used there to provide end-to-end guarantees for robustly learning multi-index models.

3. What is the main technical difficulty in establishing random design generalization bounds? The Rademacher complexity for the adversarial loss has been studied extensively in prior works, so this seems within reach, and with this bound we can make a more confident comparison between adversarial learning and ridge regression.

4. I couldn’t find the definition of the constant $R$ in the main text.

5. How does it make sense to talk about $\Vert d \Vert_{\mathcal{H}_j}$? Does this mean $d = \sum_j d_j$, and the norm is $\Vert d _j \Vert _{\mathcal{H}_j}$?

---
[1] A. Mousavi-Hosseini, A. Javanmard, M. A. Erdogdu. "Robust Feature Learning for Multi-Index Models in High Dimensions." ICLR 2025.

**Ethical Concerns:**

["NO or VERY MINOR ethics concerns only"]

**Final Justification:**

I didn't have any major concerns, and the other reviews are generally positive. I think the paper's approach towards relaxing the problem of robustness w.r.t. to input perturbations to robustness w.r.t. feature perturbations is interesting, which is why I recommend acceptance.

**Limitations:**

yes

**Paper Formatting Concerns:**

No concerns.

**Quality:**

3

**Strengths And Weaknesses:**

* Strengths:
The paper provides a novel and interesting trick to efficiently solve convex adversarial learning problems, and I believe there is some potential to use this insight in certain non-convex settings discussed below. The fact that the perturbation strength does not need to be adapted to the noise level makes this an appealing approach in settings where standard performance matters.

* Weaknesses:
    * In most use cases of adversarial training, one is interested in conditions of the type $\Vert x \Vert \leq \delta$ with a fixed $\delta$ to ensure worst-case input perturbations can not degrade the model performance significantly. Since the relaxation provided in this paper may not be sharp, ensuring this constraint may require a very large perturbation budget in the feature space, which would eventually hurt standard performance.
    * Some technical aspects of the paper can be improved. For example it is relatively standard to also study Rademacher-based generalization bounds in this setting, but the authors only consider a fixed-design excess risk which provides weaker guarantees and cannot rule out overfitting.
    * While the authors measure the excess risk with respect to $f^\*$, it might make more sense to compare the learned model's performance to the best model in the RKHS measured by population adversarial loss. $f^\*$ may not be a good benchmark for adversarial robustness if it changes quickly with $x$.

---

> ### Author Rebuttal · Authors · 2025-07-29
>
> Thank you very much for reading our paper and providing feedback. These are useful comments. We answer the questions **(Q)** and weaknesses **(W)** below.
>
> * **(W1) On the size of  $\\delta$ to guarantee input robustness.** Thank you for the comment. Robustness to fixed input-space perturbations is a standard goal in adversarial training. Table 1 in our paper outlines conditions under which input-space perturbations are upper-bounded by constraints in feature space. For some kernels, achieving equivalent robustness may require a relatively large feature-space radius to satisfy these conditions. We will make sure to clearly highlight this limitation in the paper. That said, the radii required to guarantee equivalence may be somewhat conservative, and our numerical experiments (Table 3) show that the method still improves robustness even when using smaller, more practical values.
> * **(W3) On optimal $f^\*$ for robustness.** In our work, we emphasize two distinct perspectives on adversarial training: generalization and robustness. While our analysis of the optimal function $f^\*$ primarily focuses on generalization, we agree that robustness is also a crucial aspect. Importantly, the two are closely connected. In particular, the expected adversarial error can be approximated as (see, e.g., \[1\]): $$\\text{Expected Adversarial Error}(f, \\delta_{\\text{test}}) \\propto \\text{Expected Error}(f) \+ \\delta\_{\\text{test}}^2 \\|f\\|\_{\\mathcal{H}}^2,$$ where $\\|f\\|\_{\\mathcal{H}}^2$ denotes the RKHS norm of the predictor. This expression shows that the regularization effect studied in our analysis also contributes to adversarial robustness at test time, with the additional term capturing the model’s sensitivity to perturbations.
>   Thank you again for the comment. We will revise the text to clarify that generalization is the primary focus of our analysis there, but also to better explain how it also provides insight into robustness.
> * **(Q1) About the comparison with input-perturbations.** Thank you for the comment. We highlight that the original problem with the perturbation in the input space is not convex. Particularly, we are considering classes of nonlinear functions where $f(x+\\Delta x)$ is often non-convex in $\\Delta x$, hence the advantage of our formulation in the feature space that yields a convex problem. While input-space adversarial training can still be approximately solved—for example, using projected gradient descent (PGD), which is common in neural network settings—we compare both approaches in our numerical experiments (see Table 3).
>
> * **(Q2) Adding relevant reference.**  Thank you. This is indeed a very relevant and interesting reference. It points to potential directions where our framework could be applied and offers valuable inspiration for future work. We will include it in the related work section and briefly discuss possible connections.
> * **(Q3 & W2) About using Rademacher complexity.** Thank you for the opportunity to clarify. In this work, we rely on Gaussian complexity ($\\gamma$) and localized Gaussian complexity ($\\beta$). Nonetheless, we agree that Rademacher complexity is a powerful tool for analyzing generalization in adversarial settings \[2\], and it could indeed be used to extend our analysis to the random design case. Below, we explain our choice of Gaussian complexity and outline how Rademacher-based extensions could be pursued.
>   1. **Gaussian vs Rademacher complexity.** Both Gaussian and Rademacher complexities are useful for studying generalization, but Gaussian complexity offers a particular advantage in regression settings: it allows the role of the noise level $\\sigma$ to appear more transparently in the analysis. In contrast, this dependence is harder to extract when using Rademacher complexity. For example, \[2, Section 7\] presents a rigorous analysis of kernel regression using Rademacher complexity, but the role of $\\sigma$ is not explicit. In our view, this transparency is one motivation for our choice. That said, Rademacher complexity could still be applied on top of our results to derive random design generalization bounds, while retaining the insights related to $\\sigma$. We consider this a natural and technically straightforward extension, and we will clarify this in Section 5.3.
>   2. **Localized Gaussian complexity.** It is also important to note that neither Gaussian nor Rademacher complexity in their standard forms yield faster rates than $1/\\sqrt{n}$. However, some of our generalization rates—such as $B^{\\text{adv}}$ and $B^{\\text{kr}}$—scale with $\\beta^2$ and can be faster, due to the use of *localized* Gaussian complexity (see, e.g., \[4\], Chapter 13). As we mention in the paper, obtaining analogous rates in the random design case would require the use of localized Rademacher complexity \[6\], which involves additional assumptions and technical challenges (see \[5\] and \[4\], Chapter 14). We believe this is a promising direction, better suited for future work.
>
>   Overall, we appreciate the suggestion. We agree that, particularly for the first case, extending our results using Rademacher complexity is both natural and feasible. We will revise Section 5.3 to include a more detailed discussion of these points and the possible extensions.
>
> * **(Q4) Definition of R.** Thank you for pointing this out. This was mentioned in the first paragraph of Section 5, $R \= \\|f^\*\\|\_{\\mathcal{H}}$ denotes the radius of the hypothesis space in the RKHS.  But we agree this can be easily missed by the reader. To improve clarity, we will add this definition directly to the relevant theorem statements to ensure they are fully self-contained.
> * **(Q5) On the definition of $d$.** Thank you very much for bringing this up. You are absolutely right that the definition given after Equation (11) is unclear and does not specify the space to which $d$ belongs.  It should be $d \= \\max\_j(d\_j)$ where $d\_j\\in  \\mathcal{H}\_j , \\|d\_j\\|\_{\\mathcal{H}\_j} \\le \\delta.$ We sincerely appreciate you spotting this issue. We emphasize that all subsequent results remain valid. We will revise the text to correct and clarify this definition.
>
> **References:**
>
> * \[1\] "Overparameterized Linear Regression Under Adversarial Attacks" A. H. Ribeiro and T. B. Schön. *IEEE Transactions on Signal Processing*, vol. 71, pp. 601–614, 2023\.
> * \[2\] "Rademacher Complexity for Adversarially Robust Generalization" D. Yin, K. Ramchandran, and P. Bartlett. *ICML*, 2019\.
> *  \[3\] "Learning Theory from First Principles" F. Bach. *The MIT Press*, 2024\.
> *  \[4\] "High-Dimensional Statistics: A Non-Asymptotic Viewpoint" M. J. Wainwright. *Cambridge University Press*, 2019\.
> *  \[5\] "Gaussian and Rademacher Complexities: Risk Bounds and Structural Results" P. L. Bartlett and S. Mendelson. *Journal of Machine Learning Research*, vol. 3, pp. 463–482, 2002\.
> *  \[6\] "Local Rademacher Complexities" P. L. Bartlett, O. Bousquet, and S. Mendelson. *Annals of Statistics*, vol. 33, no. 4, pp. 1497–1537, 2005\.

---

> > ### Comment · Reviewer_vuz8 · 2025-08-04
> >
> > Thank you for your responses. I have a follow-up question about convexity:
> >
> > I agree that the maximization is not concave in $\Delta x$, but my point is that the minimization is still convex in $f$. Notably, $\ell(y,\langle f, \phi(x)\rangle)$ is convex in $f$, thus $\max_{x \in \Omega}\ell(y,\langle f, \phi(x)\rangle)$ is convex in $f$ for any constraint set $\Omega$ since it's a maximum over convex functions. Thus the minimization of the adversarial loss can be seen as a convex problem in $f$.
> >
> > My question is that is there a chance to leverage this knowledge, or do we need both convexity in $f$ *and* concavity in $\Delta x$ for polynomial-time global guarantees?

---

> ### Author Response · Authors · 2025-08-04
>
> Thank you for the follow-up and for raising this important point. We have considered this question ourselves and thought about ***how to best leverage the structure of the problem***. As far as we can tell, the most natural approach is precisely to use adversarial perturbations in the latent space, as done in our formulation.
>
> The advantage of latent-space perturbations is that the function  $$\ell(y, \langle f, \phi(x) + p \rangle)$$ is convex in both $f$ and $p$. Convexity in $f$ follows from the fact that this is a composition of a convex function with an affine map—a  result discussed, for example, in Theorem 5.7 in Rockafellar’s “Convex Analysis”. Convexity in $p$ follows similarly, as the inner product is affine in $p$. As you correctly pointed out, the pointwise maximum of convex functions is convex (Theorem 5.5 in Rockafellar’s *"Convex Analysis"*), so ***maximizing over $p$ preserves convexity***.
>
> In contrast, the function
> $$
> f \mapsto \max_{\Delta x \in \Omega} \ell(y, \langle f, \phi(x + \Delta x) \rangle)
> $$
> is not generally convex in $f$, since $\phi(x + \Delta x)$ is nonlinear in $\Delta x$, and the function $ \Delta x \mapsto \langle f, \phi(x + \Delta x) \rangle $ is not affine unless $\phi$ is affine. Therefore, the conditions used in the latent space case do not apply here.
>
> Concavity indeed could allow us to use another set of classical results, but this would not work for general nonlinear maps $\phi$. Overall, our understanding is that the ***convexity of the outer problem cannot be guaranteed in the input disturbance setting without some form of relaxation***. This is why we believe that applying perturbations directly in the latent space offers quite a natural way to leverage the structure in this setting.
>
> **Reference:**
> - Rockafellar, R. (1970). *"Convex Analysis"*. Princeton Mathematical Series, 28.

---

> ### Comment · Reviewer_vuz8 · 2025-08-05
>
> Thanks for your response. However, I think
> $$f \mapsto \max_{\Delta x \in \Omega} \ell(y, \langle f, \phi(x + \Delta x)\rangle$$
> is in fact convex for *any feature map* $\phi(x)$. For a fixed $x,y$, define $\ell_{\Delta x}(f) = \ell(y, \langle f, \phi(x + \Delta x)\rangle)$. Then, the above is just
> $$f \mapsto \max_{\Delta x \in \Omega} \ell_{\Delta x}(f)$$
> Note that each $f \mapsto \ell_{\Delta x}(f)$ is convex, therefore their point-wise maximum is also convex. Specifically,
> for any $t \in [0,1]$, $\Delta x \in \Omega$, and $f_1,f_2$,
> $$\ell_{\Delta x}(tf_1 + (1-t)f_2) \leq t\ell_{\Delta x}(f_1) + (1-t)\ell_{\Delta x}(f_2) \leq t\max_{\Delta x \in \Omega}\ell_{\Delta_x}(f_1) + (1-t)\max_{\Delta x}\ell_{\Delta x}(f_2).$$
> Taking maximum on the LHS,
> $$\max_{\Delta x \in \Omega}\ell_{\Delta x}(tf_1 + (1-t)f_2) \leq t\max_{\Delta x}\ell_{\Delta x}(f_1) + (1-t)\max_{\Delta x}\ell_{\Delta x}(f_2).$$
> This doesn't seem to depend on any property of the feature map $x \mapsto \phi(x)$. However, what I believe is plausible is that the convexity of the outer function does not lead to a useful optimization algorithm since the function is still defined in terms of an inner maximization which is not concave, so it's not clear how to efficiently solve it.
>
> Please let me know if I missed something.

---

> ### Author Response · Authors · 2025-08-05
>
> Thank you for elaborating on it—this was helpful!!
>
> Indeed, you are correct about the convexity of the outer function. That said, as you pointed out, the non-concavity of the inner maximization still poses practical challenges for optimization. So even though the outer objective is convex, it's not clear how to efficiently solve the overall problem, at least not in an obvious way. That said, it’s certainly an interesting point and it could indeed be a future direction to look into.

---

> > ### Comment · Reviewer_vuz8 · 2025-08-05
> >
> > Thanks for your explanation. I don't have any major concerns, and I'll increase my score.

---

> > > ### Author Response · Authors · 2025-08-06
> > >
> > > Thank you very much for that, and for all the interesting comments. We really appreciate it.

---

### Official Review · Reviewer_zxfo · 2025-07-01

**Clarity:** 3
**Significance:** 2
**Originality:** 2
**Rating:** 3
**Confidence:** 3

**Summary:**

This paper proposes an adversarial training framework used to kernel-based linear regression models, aiming to enhance model robustness. Traditional adversarial training methods typically formulate robustness as a min-max optimization problem in the input space and solve it using computationally intensive projected gradient descent (PGD). In contrast, this study introduces perturbations directly in the reproducing kernel Hilbert space (RKHS) feature space rather than the input space, enabling a convex reformulation of the problem that admits a closed-form solution. An optimization algorithm based on iterative kernel ridge regression is proposed. The paper establishes the convexity of the proposed formulation and analyzes its theoretical properties, deriving generalization bounds that guarantee its performance. Furthermore, the methodology is extended to multiple kernel learning, thereby increasing model expressiveness. Experimental evaluations on synthetic and real-world datasets demonstrate the superior performance of the proposed method compared to traditional kernel ridge regression approaches.

**Questions:**

● Can the proposed adversarial training framework be effectively scaled to handle larger datasets?
● Is it possible to extend or adapt this method for classification tasks, and what modifications would be necessary?
● How does the computational complexity of the proposed method compare with existing large-scale kernel methods?
● Could alternative kernel approximation methods or matrix decomposition techniques address the computational complexity issue effectively?
● Are there practical scenarios or applications where the feature-space perturbation provides significantly better robustness compared to input-space perturbations?
● Are you perhaps confusing excess risk(used in line 150) with the L2 error? Excess risk refers to the difference between the risk of the empirical risk minimizer (ERM) and the risk of the Bayes estimator, and it is conceptually distinct from the L2 error.
● In Table 3, what does the parameter p represent? In Appendix Table S.2, p appears to refer to the p-norm, whereas in Appendix E.1 and Table 2, p is used to denote the number of features in the data. This inconsistent notation is quite confusing. Moreover, the role of the p-norm is not clearly explained in the paper.
● Traditional kernel regression already achieves robustness through the use of a ridge penalty, so it is unclear what additional benefits your method provides. Are there specific scenarios where standard kernel ridge regression is not robust, but your approach demonstrates robustness?

**Ethical Concerns:**

["NO or VERY MINOR ethics concerns only"]

**Final Justification:**

The authors propose a method that applies adversarial learning in the feature space as a way to fit robust models. However, it remains unclear how much advantage this approach provides over standard kernel ridge regression. To properly assess the merit of the proposed method, experiments on larger-scale tasks. Such as high-dimensional image classification or applications involving large language models (LLMs) would be necessary, in addition to the small to medium scale regression problems currently presented in the paper.

**Limitations:**

● The proposed framework does not address the high computational complexity of kernel regression, which involves O(n³) operations for kernel matrix inversion.
● The methodology and experiments are restricted to regression problems, without considering or testing its generalization to classification tasks.
● The experimental validations provided are limited to relatively small datasets, raising questions about the scalability and numerical efficiency of the approach.

**Paper Formatting Concerns:**

Upon inspection, I found no violations of the NeurIPS 2025 formatting guidelines.
The page count is within the 9‑page limit, margins and fonts conform to the specification, the title and section headings are properly styled, figures and tables are legible and captioned correctly, the checklist appears after the references, and no author information is present—everything aligns with the requirements.

**Quality:**

2

**Strengths And Weaknesses:**

Strengths:

● Introduces perturbations in the RKHS feature space rather than the input space during adversarial training, presenting a closed-form solution for the inner-maximization problem, thus eliminating the need for PGD.
● Provides theoretical insights by deriving upper bounds on the empirical L2 prediction error associated with the solutions obtained by the proposed algorithm, offering theoretical guarantees for performance.


Weaknesses:

● Kernel regression inherently involves computing the inverse of the kernel matrix, resulting in a high computational complexity of O(n³). Although acknowledged by the authors, this limitation is not addressed within the scope of the paper.
● The proposed framework is limited to regression problems, lacking discussion or experimental validation for classification scenarios.
● Due to methodological constraints, the diversity of experiments is limited. The datasets employed in experiments have relatively small numbers of features and data points, raising concerns about the scalability and numerical efficiency of the proposed method.
● Additionally, the paper does not include a proof for Proposition 1, which is central to the theoretical foundation of the proposed method.

---

> ### Author Rebuttal · Authors · 2025-07-29
>
> Thank you very much for reading our paper and providing feedback. These are very useful comments. We answer to the  questions and concerns below:
>
> * **Scalability and numerical efficiency.** In this paper, we focus on medium-scale problems, which allows us to include generalization analysis and multiple kernel learning within the scope of the study. Nonetheless, the method is well-suited for scaling to larger settings. The main computational bottleneck—solving a reweighted ridge regression—can be solved approximately, for instance by using a small number of iterations of a first-order method, thus avoiding the $\\mathcal{O}(n^3)$ complexity of exact solvers. Ribeiro et al. \[1\] show that a conjugate gradient (CG)–based approach can replace costly matrix decompositions such as Cholesky or SVD. This significantly reduces the per-iteration cost and makes the adversarial training framework computationally feasible for large-scale applications.
>
>  * **Usage of large-scale kernel methods and approximations.** We also highlight that our method solves a sequence of reweighted kernel ridge regression problems, hence it can naturally benefit from the same computational accelerations developed for large-scale kernel methods. Hence that any scalable solver for kernel ridge regression—such as those based on random feature approximations, Nyström methods, or iterative solvers—can be directly integrated as a subroutine in our algorithm. We will clarify this point in the revised manuscript.
>
> * **Classification.** Thank you for the question. This is a valid and important point, and indeed one we discussed internally. We chose to focus on regression in this paper to keep the presentation clear and focused. Regression offers a rich and well-understood setting that allows us to highlight the core ideas and trade-offs of adversarial kernel learning. That said, many of our results can be extended to classification. The main idea—adversarial training in feature space—is applicable to both regression and classification tasks. Proposition 1, which connects input- and feature-space perturbations, holds for classification. Proposition 2 also has a natural analogue under similar regularity conditions. The optimization algorithm we propose is tailored to the square loss, but related approaches for classification exist. Indeed, Ribeiro et al. \[1\] provide an algorithm in the linear setting that could be adapted to the nonparametric case considered here. Sections 5 and 6, which deal with generalization and empirical performance, could in principle be adapted for classification, but doing so would require additional theoretical and experimental work. We will revise the text to clarify these points and highlight potential directions for extension.
>
>
>
> * **Scenarios where the feature-space perturbation provides better robustness than input-space perturbations.** Thank you for the question. Interestingly, in several scenarios, we observe that adversarial training in feature space leads to improved robustness compared to direct input-space perturbations. This is illustrated in Table 3, and we will expand the paper to include additional examples for a more direct comparison.
> Below, we present a representative example using the pollution dataset.  We report results under a standard threat model with $\\ell\_\\infty$-bounded input perturbations of radius $\\delta\_{\\text{inp}} \= 0.1$ . We include the results for the two values of $\\delta$, illustrating the trade-off between clean and robust accuracy: increasing the feature-space radius $\\delta$ enhances robustness, but comes at the cost of slightly reduced clean performance. Notably, in both cases, our method outperforms input-space adversarial training—even when the latter uses the same threat model as the one applied during evaluation. We believe this is due to the fact that our method enables exact optimization of the worst-case feature-space loss, whereas input-space adversarial training typically relies on approximate solvers such as PGD.  We will add this comparison and discussion to the paper to highlight scenarios where our method achieves better robustness than traditional input-space adversarial training. *We show the $R^2$ score (higher is better).*
> | $R^2$ score  | Clean data  |  Adversarial attack  |
>  | :---- | ----: | ----: |
> | **Adv Kern** (ours, $\\delta\=0.01$) | **0.72** | 0.51 |
> | **Adv Kern** (ours, $\\delta\=0.1$) | 0.67 | **0.58** |
> | **Adv Inp** (same threat model as evaluation) |  0.68 | 0.50 |
>
>
>
> * **Excess risk**. Thank you for bringing this up.  We agree that this could cause confusion; we will revise the text to better clarify why we refer to this quantity as the *excess risk*. We highlight, though, that our definition does follow from what you said. Particularly,  $$\\frac{1}{n}\\sum\_{i=1}^n (f(x\_i) \- f^\*(x\_i))^2  \= \\text{Expected risk} \- \\text{Bayes risk} $$  Where$$\\begin{aligned}  \\text{Bayes risk}& \= \\mathbb{E}\_{w}\[\\frac{1}{n}\\sum\_{i=1}^n (y \- f^\*(x))^2\]  \= \\mathbb{E}\_{w}\[\\frac{1}{n}\\sum\_{i=1}^n (\\sigma w\_i)^2 \]  \= \\sigma^2\\\\
>   \\text{Expected risk}&=  \\mathbb{E}\_{w}\[\\frac{1}{n}\\sum\_{i=1}^n (y \- f(x))^2\]  \= \\mathbb{E}\_{w}\[\\frac{1}{n}\\sum\_{i=1}^n (\\sigma w\_i \+ f(x\_i) \- f^\*(x\_i))^2 \] \= \\sigma^2 \+ \\frac{1}{n}\\sum\_{i=1}^n (f(x\_i) \- f^\*(x\_i))^2.
>   \\end{aligned}$$ We will ensure this reasoning is made explicit and clearly presented in the revised manuscript.
> * **Parameter $p$ and inconsistencies in the notation.** Thank you a lot for catching this up. Indeed, we use the parameter $p$ to refer to the $p$-norm in Table 2 and to the number of parameters in Table 2\. We agree that this could cause confusion to the reader and we will fix it in the revision.
> * **Improved robustness when compared to kernel ridge regression.**  While kernel ridge regression already offers some robustness via the ridge penalty, our method explicitly minimizes the worst-case loss over a feature-space perturbation set (Eq. 2), directly targeting adversarial robustness. This results in *better defense against adversarial disturbances in all the cases we evaluated*, with our method outperforming kernel ridge regression under input-space attacks (see Table 3 and Figure 3). This demonstrates the practical robustness benefits of our formulation beyond those offered by standard ridge penalties.
> * **Proof of proposition 1\.** Thank you for the comment.  Proof of proposition 1 is given in the Appendix (A.1). We will add a pointer to it in the main text.
>
> **References**
>
> - \[1\] "Efficient Optimization Algorithms for Linear Adversarial Training" A. H. Ribeiro, T. B. Schön, D. Zachariah, and F. Bach. AISTATS, 2025\.

---

> > ### Comment · Reviewer_zxfo · 2025-08-06
> > **Official Comment by Reviewer zxfo**
> >
> > Dear Authors,
> >
> > Thank you for addressing my questions. Most of my concerns have been resolved. However, I still find it difficult to gauge whether your method yields a improvement over kernel ridge regression. While Table 3 reports some gains, the performance differences in Figures 2 and 3 appear marginal. Even accounting for the fact that Figure 2 omits adversarial attacks, it would strengthen your case to include additional experiments on real-world datasets.
> >
> > Finally, I look forward to seeing results on large-scale datasets in the revised version to demonstrate the method’s scalability and robustness.
> >
> > Best regards.

---

> ### Author Response · Authors · 2025-08-06
>
> Thank you very much for the follow-up question. We agree that additional experiments on real-world and large-scale datasets could further strengthen the empirical evaluation. Some real-world datasets are already included, and we will broaden the evaluation in the revision of this work.
>
> About the comparison with kernel ridge regression. There are essentially two main advantages. The ***first is robustness to adversarial attacks***, as illustrated in the last columns of Table 3: our method provides improved performance under adversarial perturbations.
>
> The ***second is generalization using a hyperparameter that does not depend on the noise level***. This is the focus of Figures 2 and 3. The comparison in these figures highlights this aspect, and not necessarily performance gains. Here, kernel ridge regression uses ***cross-validation*** to select hyperparameters, whereas our method achieves competitive performance using a ***default parameter derived directly from the problem dimensions***. This represents a practical advantage: strong performance without the need for tuning, which is particularly valuable when cross-validation is computationally expensive or when data is limited. The property is also supported by the generalization analysis provided in Section 5.

---

### Official Review · Reviewer_pZHY · 2025-07-02

**Clarity:** 3
**Significance:** 3
**Originality:** 3
**Rating:** 5
**Confidence:** 3

**Summary:**

This paper establishes the theoretical framework for kernel learning in the context of adversarial training and RKHS.
The proposed framework handles feature-space perturbation.
The efficient algorithm for solving the kernel adversarial training has been proposed, and generalization properties for the proposed framework are also provided.

**Questions:**

1. If \delta follows different orders of the sample size, then what is the order of the generalization bound?
2. How do the authors compare their results for kernel ridge regression in Theorem 2 with the rates developed in traditional statistics?

**Ethical Concerns:**

["NO or VERY MINOR ethics concerns only"]

**Final Justification:**

I believe this paper has made solid contributions in establishing a framework for kernel learning in the context of adversarial training and RKHS. Also, the authors have given the dependence of generalization bounds on the sample size and provided some comparisons with kernel ridge regression. I am satisfied by the authors' response.

**Limitations:**

Yes

**Quality:**

3

**Strengths And Weaknesses:**

Strengths
1. This paper is well-written and solid.
3. This paper provides a new theoretical perspective of adversarial training from function spaces and non-parametric models.
4. The authors have considered different aspects of the proposed kernel adversarial training (formulation, numerical optimization, generalization bounds, multiple kernel learning, etc.)


Weakness
1. It will be better if the authors can explain how the generalization bound of the kernel adversarial training relies on the sample size n under different settings of the upper bound of adversarial error and compare this with kernel ridge regression and original adversarial training.
2. Some of the theorems/propositions in this paper are not self-contained. For example, what are R,\lambda, \sigma in Theorem 1 and Theorem 2? I would appreciate it if the authors could adjust them.
3. UpdateWeights in Algorithm 1 is confusing. Could the author write the details of this step?

---

> ### Author Rebuttal · Authors · 2025-07-29
>
> We greatly appreciate the reviewer’s time and attention in evaluating our research. We are happy to receive an overall positive feedback and also comments that can help us to improve clarity. We respond to the Questions **(Q)** and address the Weaknesses **(W)** below:
>
> * **(Q1 & W1) Dependence of generalization bounds on the sample size.** Thank you for the comment. We agree that the presentation of our analysis in Section 5 is somewhat scattered. To improve clarity, we will add a brief summary of the key results at the beginning of the section. We obtain two rates depending on how you set $\\delta$:
>   * If $\\delta\\propto \\gamma$  we obtain the rate $O(R\\sigma/\\sqrt{n})$ for linear, Gaussian and Mattern kernel;
>   * If $\\delta\<\\frac{\\sigma}{R} \\beta^2$ we obtain $O(\\sigma^2 p/n)$ for linear kernel, $O(\\sigma^2(log n)^c/\\sqrt{n})$  for the Gaussian kernel, and $O(\\sigma^2/n^{1/(2 \+ p/\\nu)})$ for the Matern-$\\nu$ kernel.
> * **(Q2) Comparison with kernel ridge regression.** We can set the regularization parameter to obtain the same rates for ridge regression. In the first case, however, this requires choosing $\\lambda \\propto \\sigma / (R \\gamma)$, which depends not only on the problem dimensions but also on the signal-to-noise ratio (which is typically unknown). We highlight this as a strength of Kernel Adversarial Training, which simplifies hyperparameter selection.
> * **(W2)** **Clarifying notation.** Thank you for pointing this out. The variable were defined in the first paragraph of Section 5, but we agree that the theorems should be fully self-contained.  Just for the reference $R \= \\|f^\*\\|\_{\\mathcal{H}}$ denotes the radius of the hypothesis space in the RKHS norm, $ \\lambda$ is the regularization parameter in ridge regression, and $\\sigma$ is the noise magnitude.  In the revised version, we will clearly define all quantities in the theorem statements.
> * **(W3) Clarifying step in the algorithm.**   Thank you for pointing this out\! We agree. The weight calculation is specified on the previous page (end of *4.1 Proposed Algorithm*) as
>   * $w\_i \= \\frac{1}{\\eta^{0}\_i}$, and
>   * $\\lambda \= \\frac{1}{n}\\sum\_{i} \\frac{\\delta^2}{\\eta^{1}\_i}$.
>
>   We will add this to Algorithm 1 to make it easier to read.

---

> ### Comment · Reviewer_pZHY · 2025-08-04
>
> Thanks for the authors' reply. My concern has been addressed. I will keep my score.

---

### Note · Authors · 2025-08-12

We sincerely thank all reviewers for their time, effort, and valuable feedback. We are pleased that most reviewers had an overall positive view from the start, noting that the paper is *well-written and solid* (**pZHY**), presents a *novel and interesting trick to efficiently solve convex adversarial learning* (**vuz8**), takes a *rather original approach…, well written and presented with rigor* (**PoeL**), and is *theoretically innovative* (**KwLt**).

The ***constructive criticism and comments were also highly valuable***, ***helping us identify concrete improvements*** that we will incorporate in the revision:

* **Expanded empirical analysis** – During the rebuttal, we did additional experiments that will be added to the paper.  As requested by **KwLt**, we included ***additional sensitivity analyses***. We will also report ***experiments for fixed $\\delta$***, as mentioned in our responses to **zxfo** and **PoeL**.
* **Regularization vs. robustness** – We will more clearly highlight the two distinct perspectives on adversarial training: its use to defend against perturbations in the input space, and its role as a form of regularization. This appears in the discussion with **PoeL**, **vuz8**, and **zxf**o and we believe is important clarification. ***The experiments in the response to*** **zxfo *and* PoeL *help highlight the different views.***
* **Clarifying the role of our generalization analysis**– The questions from the reviewers will be incorporated to improve notation and presentation. Particularly, we believe the exchange with **PoeL** and **vuz8** will help us refine the scope and presentation of the generalization bounds. Moreover, **pZHY**’s request for a more explicit summary will be addressed by adding a concise statement of the main results.

We are also grateful to the reviewers who very actively engaged during the rebuttal period. Several explicitly mentioned that ***their concerns had been addressed***, with some indicating they had increased their scores (**vuz8**). **PoeL** further confirmed an “*acceptance recommendation”*, which we hope reflects a move from *weak accept* to *accept*.

Overall, we believe all the changes mentioned in the rebuttal will make the final paper clearer and stronger, and that their scope is fully manageable within a standard camera-ready revision. ***We once again thank the reviewers and the Area Chair for their constructive engagement with our work.***

---

### Decision · Program_Chairs · 2025-09-17

**Decision:**

Accept (poster)

**Comment:**

This paper proposes a new adversarial training framework based on kernel space. The core idea is to transfer adversarial perturbations from the input space to a high-dimensional feature space defined by a kernel function. This conversion brings several key advantages
1. Convert the "min-max" problem that is originally difficult to optimize into a convex optimization problem, so that it can be solved efficiently.
2. The paper proves that this new objective function has a closed-form solution, which avoids iterative optimization in the inner loop.
3. This method provides an adaptive regularization effect and can achieve near-optimal performance without knowing the data noise level. This is a significant advantage compared to standard kernel ridge regression.

Overall, the reviewers think this paper is solid.